# Renewal of planktonic foraminifera diversity after the Cretaceous Paleogene mass extinction by benthic colonizers

Raphaël Morard [1] ✉, Christiane Hassenrück [1,2], Mattia Greco [3], Antonio Fernandez-Guerra [4], Sylvain Rigaud[5], Christophe J. Douady[6,7] & Michal Kucera [1]

The biotic crisis following the end-Cretaceous asteroid impact resulted in a dramatic renewal of pelagic biodiversity. Considering the severe and immediate effect of the asteroid impact on the pelagic environment, it is remarkable that some of the most affected pelagic groups, like the planktonic foraminifera, survived at all. Here we queried a surface ocean metabarcoding dataset to show that calcareous benthic foraminifera of the clade Globothalamea are able to disperse actively in the plankton, and we show using molecular clock phylogeny that the modern planktonic clades originated from different benthic ancestors that colonized the plankton after the end-Cretaceous crisis. We conclude that the diversity of planktonic foraminifera has been the result of a constant leakage of benthic foraminifera diversity into the plankton, continuously refueling the planktonic niche, and challenge the classical interpretation of the fossil record that suggests that Mesozoic planktonic foraminifera gave rise to the modern communities.

Despite the pivotal role of the fossil record of planktonic foraminifera in revealing past climates and studying plankton evolution, the origin of the group remains elusive. Conflicting evidence exists between paleontological and molecular studies on how the planktonic foraminifera emerged and diversified. The earliest record of planktonic foraminifera dates back to the early Jurassic, where their appearance is thought to have been the response to widespread oceanic anoxia[1]. The group became diverse and abundant in the early Cretaceous and in most paleontological phylogenies[2,3], the main extant clades are traced back to that time. However, molecular genetic data imply that the extant foraminifera in the plankton may be the result of repeated invasions from the benthos. This is supported by the independent colonization of the planktonic niche by the triserial *Gallietellia* during the Miocene[4] and the ongoing transition into the plankton observed in the biserial *Bolivina*[5,6]. Understanding the origin of the planktonic

foraminifera is of key importance because their stratigraphic record is used to study the interplay between diversity and climate[7] and past biological crises[8]. So far, these studies assume that the fossil record of planktonic foraminifera represents the waxing and waning of diversity generated by speciation and extinction within long-ranging clades that were able to survive past environmental crises[7]. If planktonic foraminifera represent different clades bestowed with different life traits and if the pelagic niche has been repopulated repeatedly from the benthos, the interpretation of the biotic response of planktonic foraminifera to environmental upheavals throughout the Meso- and Cenozoic would have to be fundamentally reassessed.

The fossil record of the extant foraminifera is largely represented by agglutinated and calcareous forms belonging to the clades Globothalamea, Tubothalamea, and Lagenida[9], but environmental surveys show that globally the diversity in the group is dominated by the naked

[1]MARUM Center for Marine Environmental Sciences, University of Bremen, Leobener Strasse, 28359 Bremen, Germany. [2]Leibniz Institute for Baltic Sea Research Warnemünde (IOW), Seestrasse 15, 18119 Rostock-Warnemünde, Germany. [3]Institute of Oceanology, Polish Academy of Sciences, Sopot, Poland. [4]Centre for GeoGenetics, Natural History Museum of Denmark, University of Copenhagen, Øster Voldgade 5-7, Copenhagen 1350K, Denmark. [5]52 chemin de Claret, F-05700 Serres, France. [6]Université Lyon, Université Claude Bernard Lyon 1, CNRS, UMR 5023, ENTPE, Laboratoire d'Ecologie des Hydrosystèmes Naturels et Anthropisés, Villeurbanne, France. [7]Institut Universitaire de France, 103 Boulevard Saint-Michel, 75005 Paris, France. ✉e-mail: rmorard@marum.de

Monothalamea[10] that likely emerged during the Precambrian[11]. One surprising aspect of the evolution of foraminifera is that many key innovations, such as biomineralization, emerged repeatedly and independently in the group (Fig. 1). Yet, despite the existence of the group since the Precambrian[11] and of biomineralization at least since the upper Devonian[12], foraminifera colonized the plankton only in the Jurassic and all of their planktonic representatives appear to belong to one clade of the Globothalamea, the Rotaliida. Could it be that unlike many other innovations in the group, the transition into the plankton only occurred once? With the emergence of large metabarcoding surveys[13] and improved coverage of barcode references[14], the occurrence and identity of foraminifera in the plankton can now be studied from environmental DNA sequences, which would reveal which lineages, irrespective of size and presence of shells, dwell in the plankton. Here, we show that the benthic foraminifera of the clade Globothalamea can disperse actively in the plankton and form independent holoplanktonic clades that renewed the diversity of the group in the pelagic realm after the end-Cretaceous biological crisis.

## Results and discussion
### Living benthos in the plankton
To evaluate the diversity of foraminifera in the plankton, we re-analyzed the comprehensive catalog of eukaryotic diversity of the TARA Ocean dataset[13]. We queried the environmental dataset against a reference database updated with ~2000 additional foraminifera reference sequences with curated taxonomy that include 40 of the 45 morphological species of holoplanktonic foraminifera and all main clades of benthic foraminifera, albeit with Lagenida underrepresented. We retrieved 1094 Molecular Operational Taxonomic Units (MOTUs) with at least 90% identity with the updated reference database that accounted for 1,157,287 sequences and occurred in 1002 of the

1048 samples of the TARA Ocean dataset (Supplementary Fig. 1). From those MOTUs, 346 (32%) were attributed to known holoplanktonic clades, representing 83% of the sequences (Fig. 2A). The remaining 748 MOTUs (68%) were attributed to foraminifera clades that are only known from the benthos, with the largest part belonging to the Globothalamea (416 MOTUs).

Finding DNA of benthic foraminifera in pelagic samples is not surprising. Some benthic microorganisms will always be found in the plankton because of the passive entrainment of sediment particles during storms or similar events. However, such inadvertent inhabitants of the plankton would rapidly decrease in abundance away from the source of their advection. Indeed, the likelihood of observing MOTUs of each holoplanktonic clade in our analysis does not decrease with distance from the coast, whereas the benthic Tubothalamea and "Monothalamea" OTUs always become rarer away from the coast. (Fig. 2C). Nonetheless, the "benthic" Globothalamea contain some MOTUs that also persist further off-shore and whose occurrence in the plankton thus cannot be explained by passive entrainment.

To unravel which of the Globothalamea MOTUs appeared plankton-like, we carried out the analysis of their occurrence as a function of distance from shore at the level of single MOTUs (Fig. 2D). This is a simplified model because other environmental parameters are likely to control the presence or absence of individual MOTUs, but it allowed us to differentiate MOTUs that may be passively transported compared to those capable to persist in the plankton. Indeed, the vast majority of MOTUs assigned to the holoplanktonic clades showed an increasing or constant probability of occurrence with greater distance to the coast. Inversely, the MOTUs belonging to Monothalamea and Tubothalamea mostly displayed decreasing probability of occurrence with a greater distance to the coast. Based on this observation, we used the holoplanktonic MOTUs on the one hand, and the Monothalamea and

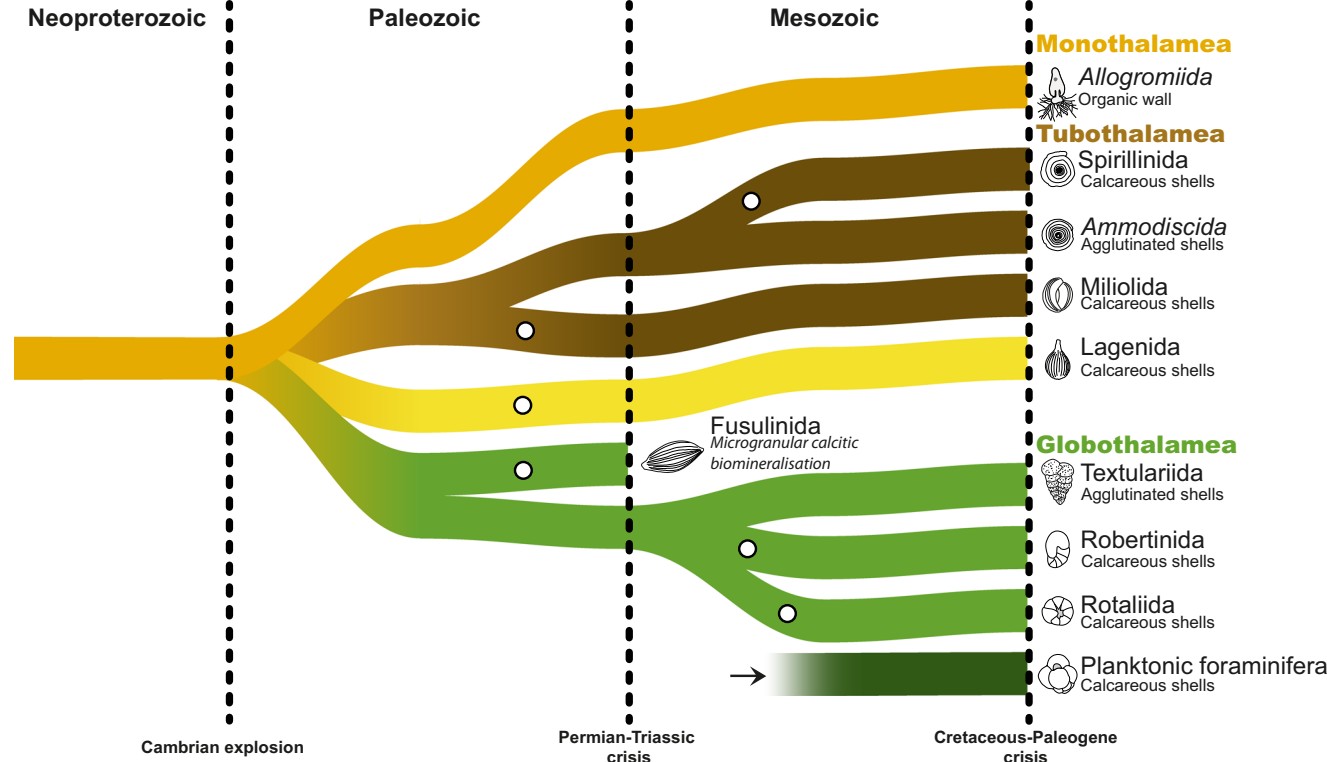

**Fig. 1 | Simplified evolutionary history of foraminifera.** The scheme depicts the diversification of foraminifera from their emergence in the Precambrian until the end of the Mesozoic with a simplified stratigraphy. The polytomies reflect uncertainties in the evolutionary relationships between the depicted lineages. The Globothalamea and Tubothalamea have a class rank and the status of the Monothalamea is uncertain. The relationship of Lagenida to other foraminifera lineages is unclear but it is a distinct lineage from the three other main clades. The open circles approximately indicate when completely biomineralized shells appeared in each lineage. The figure is based on refs. 9, 11, 57, 58.

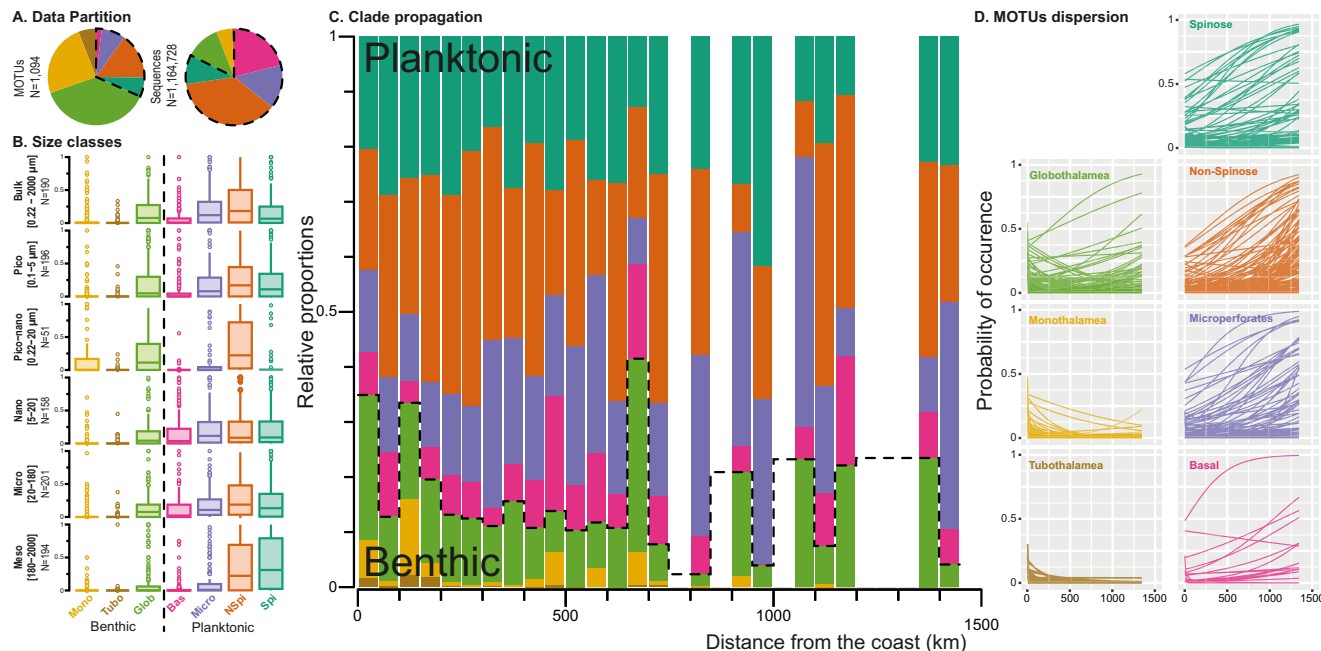

**Fig. 2 | Dispersion of benthic foraminifera in the plankton. A** Relative contribution of benthic and planktonic clades to MOTUs and sequences in the TARA Ocean dataset. **B** Sequence proportions of the main foraminifera clades across size fractions of the TARA Ocean dataset (Boxplots display the interquartile range (box), extrema (whiskers), and outliers (dots)). **C** Average sequence proportions of planktonic and benthic clades with distance from the coast in the TARA Ocean dataset. **D** Probability of occurrence of individual MOTUs of planktonic clades as a function of distance to coast based on their absence/presence in the TARA Ocean dataset. The data used to produce the figure are provided in Supplementary Data 2.

Tubothalamea on the other hand, to train two complementary supervised Random Forest models to classify the Globothalamea MOTUs as either "planktonic" or "benthic" according to (i) their presence/absence patterns and (ii) probability of occurrence (Fig. 3A). The Random Forest models performed with an accuracy of 94.4% (i) and 96.9% (ii) associated with an F1 score of 0.8 (i) and 0.9 (ii), respectively, and were in agreement in 71% of the predictions, representing 46 Globothalamea MOTUs that were considered as "putative planktonic", i.e., showing a pattern of occurrence in the plankton, which is consistent with the pattern of occurrence of planktonic taxa.

To determine the taxonomic affiliation of the Globothalamea MOTUs found in the plankton, we produced a backbone phylogeny of Globothalamea that encompassed all documented families of the rotaliids[15] and mapped the MOTUs using a phylogenetic placement approach (Fig. 3B). This analysis revealed that the ability to persist in the plankton is widespread among the Globothalamea and should therefore be considered a synapomorphy of the clade. The occurrence of the "putative planktonic" Globothalamea across all size classes in the TARA dataset implies (Fig. 2B) that the ability to persist in the plankton occurs among adult individuals and is not limited to gametes or juveniles. With the exception of the tychopelagic *Bolivina*[5], actual specimens (or their shells) of the "putative planktonic" Globothalamea have only rarely been reported from the plankton[16]. This is consistent with the small share of these MOTUs among the sequences and suggests that the ability to persist in the plankton may be a dispersal strategy rather than a persistent lifestyle.

A widespread ability to disperse in the plankton, being able to remain buoyant and feed in the plankton[5] provides an obvious stepping stone on the transition from the benthos into the plankton[17]. However, this innovation was apparently in itself not sufficient to develop a full holoplanktonic lifestyle. Since among the Globothalamea, only lineages with biomineralized shells have completed the full transition to the planktonic lifestyle, the second stepping stone on the transition into the plankton may have been the ability to secrete mineralized shells. The reason why biomineralisation would be the key

to holoplanktonic lifestyle among the Globothalamea may be simple: we note that the non-biomineralising Globothalamea (textulariids) build their shells by agglutinating sediment particles and such particles are not present in the plankton, possibly preventing holoplanktonic lifestyle in the absence of complete biomineralization.

## Origin of the extant holoplanktonic foraminifera clades

The evolutionary history of planktonic foraminifera is typically presented as a narrative of a Mesozoic origin and a history of extinction and radiation events leading to their modern diversity[3,8]. This narrative is supported by the observed continuous occupation of the planktonic niche by planktonic foraminifera since their emergence in the Jurassic. It implies that the transition into the plankton occurred once, or was rare afterward, and that the main clades survived all environmental and biotic crises at least since the early Cretaceous where the fossil record is resolved, including the Cretaceous–Paleogene mass extinction[8]. Considering that the two main prerequisites for conquering the plankton, the ability to disperse in the plankton and the ability to produce biomineralized shells, were widespread among the Globothalamea, it would be surprising that this transition did not occur repeatedly.

The extant planktonic foraminifera have long been considered to represent descendants of Cretaceous lineages, with the two most diverse clades (the macroperforate Globigerinidae and Globorotaliidae) originating from among the rare survivors of the K/Pg crisis[2]. However, there exists no independent evidence for a common origin of the extant holoplanktonic clades, not even for a common origin of the two macroperforate clades. Having barcoded the entire diversity of the extant planktonic foraminifera using their SSU rRNA gene, we can now ask whether or not their molecular phylogeny supports a shared common ancestry hypothesis. To this end, we constructed a reference phylogeny using representative sequences of major benthic families of the Globothalamea, adding sequences representing all holoplanktonic foraminifera clades, but excluding long-branch taxa (see "Methods").

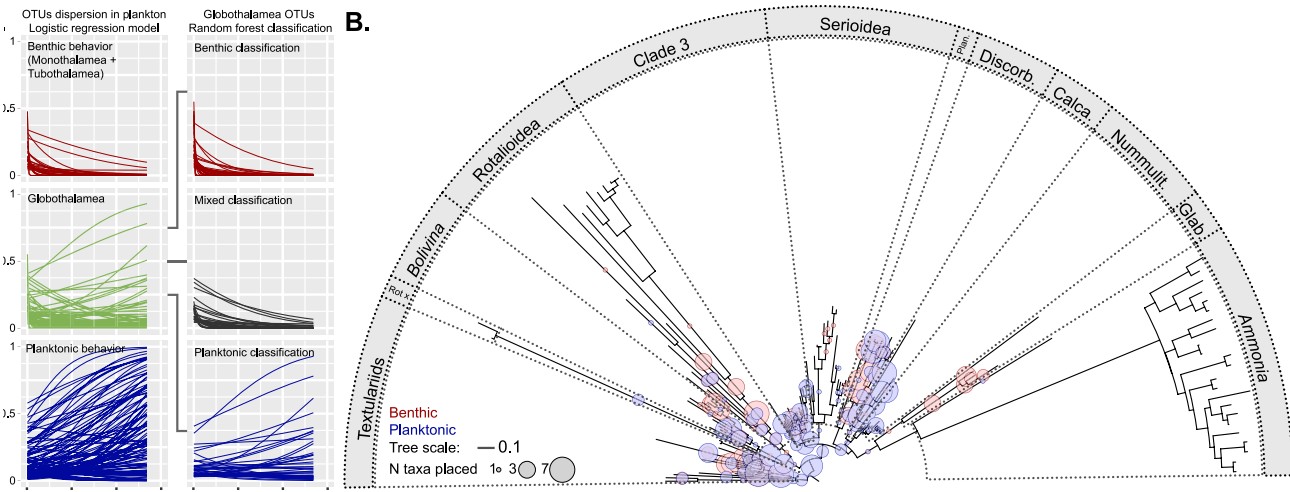

**Fig. 3 | Identification of "putative planktonic" Globothalamea. A** Random Forest classification of the Globothalamea MOTUs as either planktonic or benthic. **B** Phylogenetic placement of the MOTUs on the Globothalamea tree. The main clades of benthic foraminifera are delimited on the tree. The size of the circles represent the number of potential MOTUs affiliated with a branch. Note that the genera *Ammonia* and *Bolivina* are branching out of their home clades (Serioidea and Rotalioidea). Rot X = Rotaliida X, Plan. = Planorbulinoidea, Calc. = Calcarinoidea, Nummulit. = Nummulitoidea, Glab. = Glabratelloidea. The data used to produce the figure are provided in Supplementary Data 2 for panel **A** and Supplementary Data 3 for panel **B**.

The phylogenetic inferences (Fig. 4) revealed topologies with all Globothalamea families resolved and all three main holoplanktonic clades being monophyletic (Supplementary Fig. 2). The holoplanktonic clades, however, branch in different parts of the tree, implying at least five independent origins of the extant holoplanktonic foraminifera. This topology is consistent with earlier molecular phylogenies[18], but since the bootstrap support is not sufficient to robustly resolve their phylogenetic relationships, we subsequently formally tested two alternative hypotheses for the origin of the extant planktonic foraminifera. A Swofford Olsen Waddell Hillis test implemented in SOWHAT[19] confirmed that it is unlikely that all the extant planktonic foraminifera are monophyletic ($P = 0.001$), or that even the two macroperforate lineages are monophyletic ($P = 0.003$). A similar result is obtained using the Approximately Unbiased (AU) test[20], which rejected the monophyly of all clades ($P < 0.05$ irrespective of internal topology, Supplementary Fig. 3) and provided no support for monophyly of the two macroperforate clades ($P = 0.15$ against $P = 0.91$ for the topology in Fig. 4). This would explain why the two main clades of planktonic foraminifera that diversified after the K-Pg boundary have stark morphological differences (with and without spines), which was difficult to reconcile with a common ancestry. Clearly, a complete or partial monophyletic origin of the holoplanktonic clades is not compatible with the molecular dataset and the extant diversity of planktonic foraminifera is therefore more likely the result of multiple independent invasions of the plankton from different benthic ancestors.

Next, we can use phylogenetic inference to estimate the time of divergence of each of the extant planktonic clades from their nearest extant benthic relative. Because the foraminifera has different rates of evolution[21], we used a relaxed clock model and applied multiple dates to calibrate the molecular tree. Since the topologies of the dated trees are not supported, we cannot assume that the nodes between the sister benthic and planktonic clades represent their closest, thus youngest, last common ancestry. However, since there is no evidence for any of the known fossil or extant clades of planktonic foraminifera to have returned back to the benthos, the last common ancestor of an extant planktonic lineage and any relative from among the extant benthic foraminifera must have been benthic and therefore the age of

the inferred divergence is informative, irrespective of the support for the topology. In fact, it is important to stress that the resulting divergence age estimates provide the maximum ages of a transition into the plankton in each clade. This is because the earliest representative of a lineage leading to the planktonic clade could have still been benthic and the transition into the plankton may have occurred later. Here we retain the divergence age from the nearest benthic ancestor as a conservative estimate of the benthic-planktonic transition and use these conservative estimates to discuss the compatibility between fossil and molecular evidence concerning the evolution of the modern planktonic foraminifera (see Fig. 4 and "Methods" for details). The results reveal that the mean benthic divergence age estimates for the four main clades all cluster around the K/Pg boundary, and that the most recently diverged *D. anfracta* and *G. vivans* very likely colonized the plankton later, during the last 30 Ma of the Cenozoic (Fig. 5). The time-calibrated phylogenies based on the maximum likelihood and Bayesian inferences returned essentially the same results (Supplementary Data 5). The large uncertainties on the divergence age estimates reflect the heterogeneity in substitution rates, and are a common feature for time-calibrated trees even for inferences based on phylogenomic datasets that have perfect branch support and include sequences from hundreds of genes[22]. However, even with these large uncertainties, the divergence age estimates for all four main extant clades are incompatible with their origin from the Cretaceous planktonic foraminifera.

If the extant clades originated among the Cretaceous planktonic foraminifera, their divergence from nearest benthic ancestors must have occurred before the planktonic lineages emerged in the fossil record. Since the fossil record of Cretaceous planktonic foraminifera is well documented[3], and it is well known that these lineages have been planktonic throughout their existence[23], their divergence from the benthos must have occurred in the Early Cretaceous at the latest. Specifically, the macroperforate Cenozoic planktonic foraminifera are considered to have originated from two species of *Hedbergella*, which appear to have survived the K/Pg crisis[24]. However, the *Hedbergella* lineage can be traced in the fossil record to 140 Ma, implying a divergence age from the benthos, which is older than any of the molecular clock estimates for the benthic divergence of the extant taxa, even

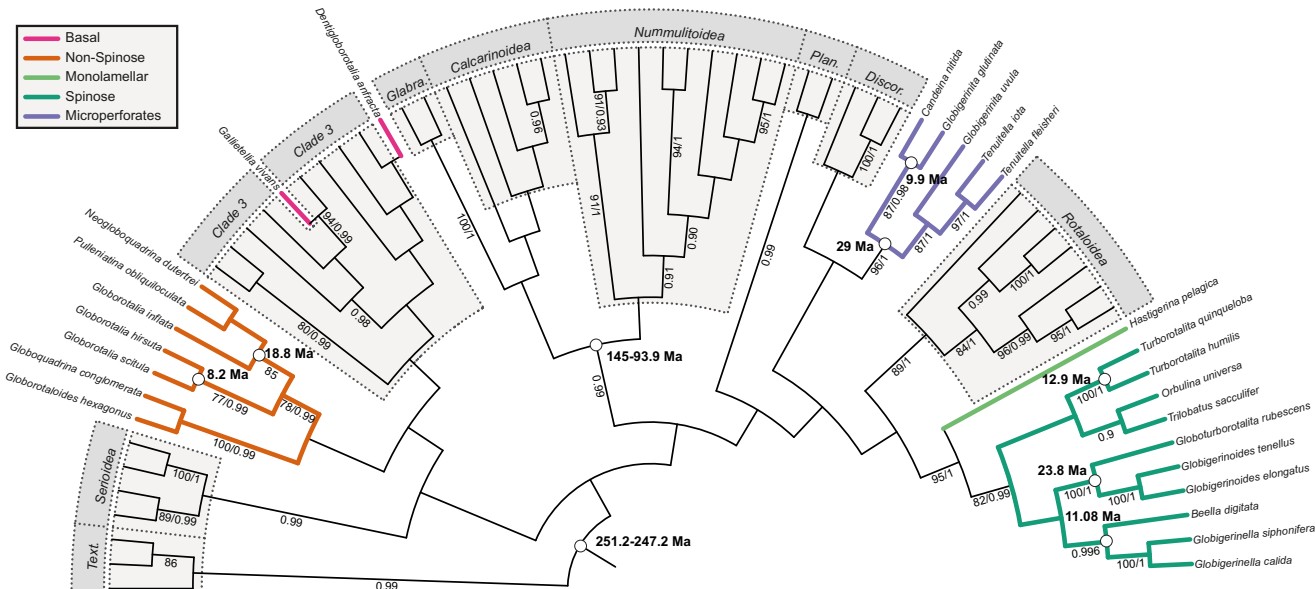

**Fig. 4 | Phylogenetic relationship of extant benthic Globothalamea with planktonic clades.** RAxML consensus topology showing the evolutionary relationships between benthic Globothalamea clades (gray boxes) and the planktonic clades (colored lines). The tree is based on 44 benthic and 25 planktonic foraminifera sequences. Bootstrap and posterior probability values above 80% or 0.9 are indicated next to the branches. Dates used to calibrate the molecular clock are indicated next to the nodes; see "Methods" for details of the calibration dates. The tree is rooted on the Textulariida. The RAxML and Bayesian topologies with the branch lengths are provided as Supplementary Fig. 2 and all relevant data files are provided in Supplementary Data 4.

when the large age uncertainties are considered (Fig. 5). Therefore, the molecular clock estimates are inconsistent with a descent of the modern macroperforate foraminifera from the *Hedbergella* lineage and in a similar manner from any other candidate ancestral lineage of Cretaceous planktonic foraminifera.

Instead, the inferred chronology of the emergence of the extant holoplanktonic clades implies that the foraminifera colonized the planktonic niche repeatedly and that different groups of the calcifying Globothalamea gave rise to the extant clades. It also implies that the Cretaceous planktonic foraminifera community did not leave any descendants surviving to the present. This means that although some Cretaceous planktonic foraminifera may have survived the crisis, such as *Guembelina* and *Muricohedbergella*, these lineages were less successful in populating the pelagic habitat than the clades newly emerging from the benthos. This conclusion is consistent with the interpretation of the post-K/Pg fossil record by refs. 25, 26, who questioned the postulated continuity between the Cretaceous *Muricohedbergella* and the earliest Cenozoic macroperforate planktonic foraminifera. Furthermore, ref. 27 suggests that the Microperforates clade diverged from a benthic ancestor (*Praepararotalia*) during the Lutecian (47.8–41.2 Ma) which is in the range of our estimation. Thus, in the scenario implied by our molecular clock estimates, the few Cretaceous planktonic foraminifera species that survived the crisis and their descendant did not gave rise to the modern communities during the Paleogene and the planktonic niche has been colonized from among diverse benthic Globothalamea which survived the crisis relatively unscathed[28,29] and renewed the foraminifera pelagic diversity.

**Consequences for the interpretation of the fossil record**

In our inferred evolutionary model, the ability to invade the pelagic habitat from the benthos emerged in the foraminifera because of the existence of a pelagic dispersal and biomineralized shells. This model provides an explanation for several unresolved questions regarding the emergence and evolutionary history of planktonic foraminifera. The observation that pelagic dispersal is limited to the Globothalamea and that the holoplanktonic lifestyle has only been adopted by the Rotaliida explains the timing of the emergence of the first planktonic

foraminifera in the Jurassic[1] as a consequence of the radiation of the Globothalamea following the Permo–Triassic mass extinction and biotic exchange. The evidence for repeated colonization of the plankton explains the apparent phylogenetic discontinuity among the earliest Jurassic planktonic foraminifera, which have aragonitic shells and may have originated from a different lineage[30,31], and the Cretaceous clades, as well as the emergence of the conspicuous Cretaceous biserial and multiserial forms[3] as the results of multiple independent colonization events. It also implies that the extent of the K/Pg extinction among planktonic foraminifera was larger than previously thought and it is no longer necessary to find explanations on how some species could survive in the plankton in the wake of the extinction event, when the pelagic food chain broke down and primary production was virtually halted[32]. The repeated seeding of the pelagic niche from benthic ancestors appears to be a continuous process that has also taken place outside of the mass extinction intervals, as documented by the young divergence ages of *Dentigloborotalia* and *Gallitellia*[4]. This observation implies that the evolutionary history of planktonic foraminifera cannot be interpreted as an extinction-speciation process acting within a single clade and that this conclusion applies even to the dominant macroperforate clades[7]. Instead, diversity among the planktonic lineages can also be generated by colonization from unrelated benthic lineages. Clearly, the interpretation of the fossil record of planktonic foraminifera biodiversity, a key testing ground for macroevolutionary models, requires a fundamentally different approach.

## Methods

### TARA Ocean dataset re-classification

We downloaded the entire V9 TARA Ocean dataset and associated metadata available at https://doi.org/10.5281/zenodo.3768509 and https://doi.org/10.1594/PANGAEA.875577 that included 1,775,314,734 sequences represented in 474,303 MOTUs from 1046 samples collected at 189 sampling stations[13] mostly in the upper 100 m of the water column. To retrieve the all foraminifera MOTUs, we updated the reference database PR²_V9 (https://zenodo.org/record/3768951) used for the assignment of the environmental metabarcodes. We removed the 927 reference

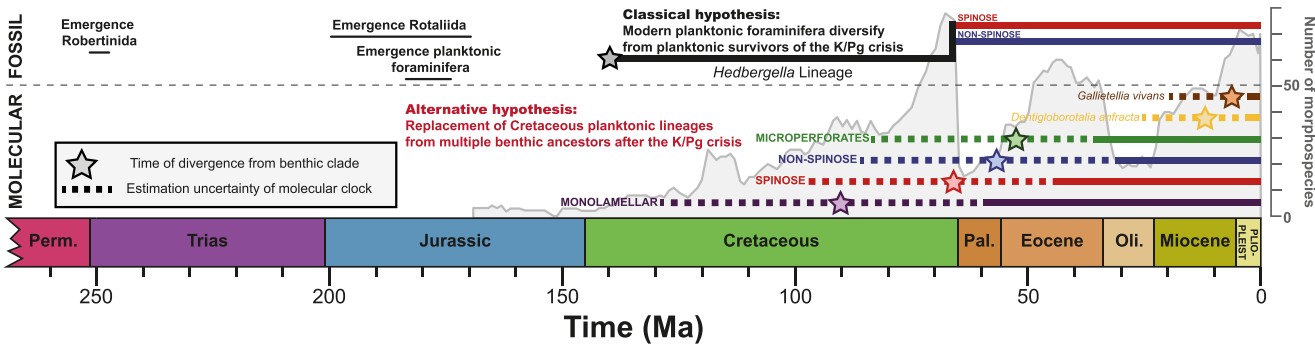

**Fig. 5 | Emergence of modern planktonic foraminifera.** The emergence of the modern clades of planktonic foraminifera in relation to the evolution of their diversity through the Mesozoic and Cenozoic following the classical hypothesis based on fossil record data and our alternative hypothesis based on molecular data. The gray curve represents the number of planktonic species observed in the fossil record (from ref. 8). The time of divergence between the extant clades and their nearest benthic relative in the maximum likelihood molecular phylogeny is represented by a star. It represents the earliest possible date of transition into the plankton. Dashed lines represent the uncertainty of the time of divergence. Time range of evolutionary events in the Mesozoic are provided above the graph. The molecular clocked phylogenies are provided as Supplementary Data 5.

sequences labeled as Foraminifera in the PR²_V9 database and replaced them with 3390 foraminifera barcode sequences with updated taxonomy following the 10 rank system of the PR² database[14]. The database then included 2044 planktonic foraminifera sequences from the PFR² database[33] and sequences published in recent work[34,35], that are organized in 4 clades (Spinose, Non-Spinose, Microperforates, Basal) and cover 40 of the ~45 species described by morphotaxonomy. The benthic foraminifera sequences are classified in the three major clades Monothalamea, Tubothalamea, and Globothalamea after ref. 9 and the taxonomy of the Globothalamea has been further updated after ref. 15. The updated V9 reference for foraminifera includes 496 unique taxonomic paths and is provided as Supplementary Data 1.

We re-classified the TARA Ocean V9 MOTUs using blast[36] v. 2.7.1 against the updated reference database. We retrieved all sequences with a percentage of identity of 90% or more to the reference for further downstream analysis. Based on the taxonomic affiliation returned by BLAST, we classified each MOTUs into four categories for planktonic foraminifera: "Spinose", "Non-Spinose", "Microperforates" and "Basal", and three categories of Benthic foraminifera: "Tubothalamea", "Monothalamea" and "Globothalamea". The results of the assignment are shown in Supplementary Fig. 1 and the occurrence of foraminifera MOTUs with the updated assignment is available in Supplementary Data 2.

### Dispersion of benthic OTUs in the plankton

Based on the updated taxonomic assignment, we calculated the portion of the diversity (percentage of MOTUs) and the volume of the data attributed to each clade (Fig. 2A), their distribution in each size class of the TARA Ocean dataset (Fig. 2B), the average proportion of each clade against the distance to the coast (Fig. 2C), and the probability of occurrence of individual MOTUs based on their presence/absence at each station using logistic regression (Generalized Linear Model implemented in R v. 4.0.2[37]; Fig. 2D). To predict the lifestyle of the Globothalamea OTUs as either planktonic or benthic, we pursued a dual approach. We applied random forest models using (1) the coefficients of the logistic regression models, implemented in the package randomForest[38], and (2) the presence/absence MOTU table directly, implemented in the mlr3 package[39]. Only MOTUs occurring in at least five stations were considered for this approach. The random forest models were trained and validated with 112 MOTUs affiliated with holoplanktonic foraminifera lineages ("Planktonic") and 32 MOTUs affiliated with the monothalamids and Tubothalamea clades ("Benthic"), to predict the ecology of 126 Globothalamea MOTUs. To reduce the risk of false positives, we only considered the Globothalamea MOTUs that were predicted as putative planktonic and benthic by both approaches for further analyses. The results of the classification approaches are provided as part of Supplementary Data 2 and the analysis code is provided on Github (https://github.com/chassenr/ForamsOrigin) and Zenodo (https://doi.org/10.5281/zenodo.7274980).

### Phylogenetic placement of Globothalamea MOTUs

To compare the phylogenetic affiliation of the Globothalamea foraminifera MOTUs classified with either a "planktonic" or "benthic" behavior, we relied on a phylogenetic placement approach. We constructed an alignment that included 137 non-redundant SSU rRNA gene sequences representative of Globothalamea foraminifera diversity that covered the V9 region. We automatically aligned the sequences with MAFFT v.7[40], chose the best model of evolution according to Modeltest-ng[41] and inferred the topology using RAxML-ng[42] with 100 rapid bootstraps using the model TVM + I + G4. The returned topology was consistent with the results of ref. 15 that indicated the respective monophyly of the families within the Globothalamea. Then, we separately aligned the MOTUs classified as "Benthic" or "Planktonic" to the backbone alignment using the –add option of the online version of MAFFT. We then used phylogenetic placement EPA-ng[43] with default options and ITOL[44] to display the results (Fig. 3). The results of the automated alignments, model selection, phylogenetic inference, and EPA-ng placement are provided as Supplementary Data 3.

### Phylogenetic context of the emergence of planktonic foraminifera

We established a phylogenetic framework to contextualize the emergence of modern planktonic foraminifera. We selected 44 representative sequences of the recognized seven superfamilies of Rotaliida and the "Clade 3" as describe in ref. 15 complemented with textulariid sequences to root the phylogeny, the aim being to reconstruct all evolutionary splits leading to the present day diversity of the Rotaliida. We thus included 11 sequences belonging to the spinose and monolamelar planktonic foraminifera clades, seven sequences of the non-spinose clade, five sequences of the microperforate clade and two sequences of planktonic species with uncertain phylogenetic affiliations labeled as "Basal". We purposely excluded species with long branches to avoid artifacts and not disrupt the phylogenetic inference (*G. minuta, N. incompta, G. truncatulinoides, G. menardii, G. ungulata, G. tumida*). We used the partial fragment of ~1000 bp located at 5′-end of the 18 S because most of the planktonic foraminifera sequences are covering this

fragment only. We automatically aligned the 69 sequences with the phylogeny aware alignment method PRANK[45] that is better suited than MAFFT because of the higher heterogeneity between the sequences. The model GTR + I + G4 was selected with Modeltest-ng and the Maximum Likelihood topology was inferred using RAxML-ng with 1000 rapid bootstrapping pseudo-replicates. The bayesian inference was performed with MrBayes v.3.2.7[46] and consisted of two simultaneous chains run for 10,000,000 generations that converged with an average standard deviation of split frequencies of 0.01, and 40,000 trees there were sampled of which 10,000 were discarded as burn-in. The results of the automated alignments, model selection, and phylogenetic inferences are provided as Supplementary Data 4. The RAxML majority rule is shown as Fig. 4 with the bootstrap values and posterior probability provided next to the branches and the topologies with branch lengths are show on Supplementary Fig. 2.

Because the inferences with all planktonic clades returned a topology with each planktonic foraminifera clade being monophyletic, hence suggesting a polyphyletic origin, but with poor branch support (Fig. 4), we tested for the two hypotheses for the origin of planktonic foraminifera suggested by the fossil record. The first hypothesis postulates a single origin of the three main clades of planktonic foraminifera, consistent with the hypothesis of a single emergence in the early Jurassic[1]. We tested two topologies for this hypothesis, one where the microperforate clade would be at the base of the planktonic clade, and one topology where the spinose would be at the base. The second hypothesis assumes a common origin of the Spinose and Non-Spinose clade as descending from the *Hedbergella* genus that survived the KT crisis as suggested by ref. 2. We manually constructed the corresponding phylogenetic trees and used the Swofford Olsen Waddell Hillis test implemented in SOWHAT[19] and the Approximately Unbiased test[20] implemented in IQ-TREE2[47] to assess if these constructed topologies were potentially congruent with the dataset. The constructed topologies and the results of the SOWHAT and AU test are provided as Supplementary Fig. S3.

To estimate the time of divergence of the planktonic clades from their benthic ancestors, we applied a molecular clock estimation to the Maximum Likelihood and Bayesian topologies. Molecular clocked phylogenies aim to translate molecular distances into absolute times of divergence between the branches and rates of evolution, which requires the establishment of prior probability of distributions on parameters such as fossil calibration and branching model[48]. Bayesian molecular clock dating uses statistical distributions to characterize uncertainties in model parameters which is translated into large confidence intervals in the time divergence estimation between branches, even when using genome wide datasets[22]. The uncertainties in posterior time of divergence estimates are influenced by fossil calibration that is crucial to detect variation of the rate of evolution between species or clades. Because foraminifera display high heterogeneity in their rate of evolution, we provided the tree with multiple calibrations. Although the Globothalamea appeared during the Paleozoic[11], the fully calcified extant taxa of the group occurred in the Mesozoic after the end-Permian crisis that caused a drastic loss of foraminifera diversity and the disappearance of the largest benthic foraminifera species[49]. Right after the crisis during the Olenekian (251.2–247.2 Ma) emerged the first robertinids characterized by arago-agglutinated or mixed arago-agglutinated-secreted (with aragonite) walls[30,31] from which the robertinids developed fully mineralized tests. The emergence of the rotaliids, with fully calcitic tests is not clear. The traditional view is that the rotaliids are directly descended from the robertinids and their earliest occurrences were in the lower Jurassic (199.3–170.3 Ma) with the first observation of buliminids[50]. An alternative hypothesis is that the rotaliids evolved directly from the textulariids and

developed calcitic mineralization independently from the robertiniids[51]. We chose to be conservative and considered a divergence between the textulariids and the rotaliida during Olenekian (251.2–247.2 Ma), which is compatible with either a direct divergence from the textulariids or the robertiniids and places this split at the earliest possible date. Second, we added a constraint within the benthic Rotaliida using the divergence between the clades Nummulitoidea (planispiral) and Calcarinoidea (trochospiral). Their separation from each other and from early Rotaliidae took place in the Cretaceous[52]. The first illustrated occurrence of the calcarinoid genus *Pararotalia* being in the Cenomanian (93.9–100.5 Ma)[53], the separation of the two clades is necessarily older. To not artificially constrain the tree toward younger date estimates, we retained a potential split between these two clades in the lower Cretaceous–Cenomanian (145–93.9).

Finally and to calibrate the planktonic foraminifera, we deliberately chose split dates in the recent history of the clade because of their higher reliability and to not constrain the most internal nodes close to the divergence from benthic clades. For the Spinose clade, we used the divergence between *G. rubescens* and the genus *Globigerinoides* at 23.8 Ma[7], the divergence between *Beella digitata* and *Globigerinella* at 11.08 Ma[54], and the divergence between *Turborotalita humulis* and *Turborotalita quinqueloba* at 12.9 Ma[7]. For the Non-Spinose clade, we used the divergence time between *Globorotalia hirsuta* and *Globorotalia scitula* at 8.2 Ma, and the divergence between *Globorotalia inflata* and *Neogloboquadrina dutertrei* at 18.8 Ma[7]. Last, for the microperforate clade, we used the divergence time between *Candeina nitida* and *Globigerinita glutinata* at 9.9 Ma and the first appearance datum of *Globigerinita uvula* at 29 Ma[54].

To calculate the molecular clock phylogeny, we used a relaxed clock model implemented in BEAST v.1.8.4[55] and model parameters were set using BEAUti v1.8.4[55]. The distribution of the fixed node age prior was considered normal and the speciation rate was assumed constant under the Yule-Process. The GTR model was selected as a substitution model. A custom R script was used to derive a rooted and fully bifurcated tree from the RAxML and MrBayes returned topologies that were then used as starting tree. Markov-Chain- Monte Carlo (MCMC) analyses were conducted for 10,000,000 generations for the RAxML topology and 15,000,000 generation for the Bayesian topology, with a burn-in of 1000 generations and saving each 1000th generation. The maximum clade credibility tree with median node heights was calculated in TREEAnnotator from the BEAST package, with a burn-in of 100 trees and a posterior probability limit of 0. The resulting trees were then visualized in FigTree v. 1.3.1[56] and provided as Supplementary Data 5, and the date of divergence between the planktonic clades and their nearest neighbor in the RAxML phylogeny is shown in Fig. 5.

### Reporting summary
Further information on research design is available in the Nature Portfolio Reporting Summary linked to this article.

## Data availability
All data used in the manuscript are publically available on Zenodo (https://doi.org/10.5281/zenodo.3768509, https://zenodo.org/record/3768951) and PANGAEA (https://doi.pangaea.de/10.1594/PANGAEA.875577), the relevant intermediary files are provided as part of the Supplementary Data files. Source data for Fig. 2, consisting of occurrences of foraminifera MOTUs in Tara Ocean samples and relevant metadata of the samples, are provided in Supplementary Data 2.

## Code availability
The code used to run the random forest analyses is provided on Github https://github.com/chassenr/ForamsOrigin and Zenodo (https://doi.org/10.5281/zenodo.7274980).

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

## Acknowledgements
The project was supported by the BMBF-funded German network for bioinformatics infrastructure (de.NBI) [project ForamsOrigin] through the provision of computational resources and user support, and by the Cluster of Excellence "The Ocean Floor—Earth's Uncharted Interface" (EXC-2077, Project 390741603) funded by the German Research Foundation (DFG).

## Author contributions
R.M. and M.K. designed the study; C.H. and R.M. performed bioinformatic and statistical analyses; A.F.G. administrated the computational resources; R.M., A.F.G., C.D., and M.G. performed the phylogenetic analyses; R.M., M.K., and S.R. discussed the paleontological interpretation. R.M. and M.K. wrote the manuscript, which was edited and approved by all authors.

## Funding

## Competing interests
The authors declare no competing interests.
