## [Peer Review File · Nature Communications]

Renewal of planktonic foraminifera diversity after the K/Pg mass extinction by new benthic colonizersReviewers' Comments:

Reviewer #1:

Remarks to the Author:

Upfront I would like to highlight that I can't comment too much on the veracity of the methods, as this is not my area of expertise, I come to this story exclusively from the fossil angle and have not used any of the methods employed in this paper before.

From my perspective though, the paper is a really interesting contribution and appears to use novel methods to dig deeper into understanding the complex evolutionary history of an important fossil group, and I'm all for planktonic foraminifera evolving as indicated by this molecular data. The fact that we could explain that macroperforate species can be non-spinose or spinose because they are two distinct monophyletic clades is a super interesting result.

However, I do have a major concern about the distinction between micro- and macroperforate planktonic foraminifera throughout the MS. Much of the narrative is centred around the fact that phylogenies based on fossils consider all planktonic foraminifera to be one clade. But this is not true. The existing fossil phylogeny of Aze et al. 2011 referred to by the authors includes only the macroperforate planktonic foraminifera and by omitting the qualifier "macroperforate" when referring to these phylogenies you are consistently misrepresenting it (e.g. lines 37-38, lines 147-149, lines 157-159, lines 240 -241). It is important to make the distinction clear that Aze et al. 2011 only considered the macroperforates to be one clade and not all planktonic foraminifera. One of the main reasons the microperforates were excluded from that phylogeny was that we considered it to be true that these taxa had arisen from multiple different benthic ancestors throughout the group's history and as such could not part of what we considered to be the monophyletic macroperforates. I am happy to concede that the macroperforate could be two monophyletic groups based on this new evidence, but even if that is true, this manuscript is currently written in a way that misrepresents the previous work based exclusively on fossils.

I do have a few other comments:

1.The paper is jargon heavy. Considering Nat Comms is designed to be "wide appeal" I would think the language could do with simplifying in places. I think the MS would benefit from a better outline of what the different classes represent. I had to keep referring to the paragraph lines 49-62 to remind myself of what each name meant. A "framing" figure would be super helpful to get a better visual understanding of what the groups represent in terms of generalised ecology, phylogeny and geological history.

2.The organisation and structure are confusing in places. Lines 52 – 64 feels awkward as an end to the abstract, to the extent that I wasn't sure if the authors do think there was multiple pluses from the benthos or not? Surely the abstract should be clearer than this? Lines 123 – 136 feel out of place, it seems like discussion, but not all of the results have been presented yet. Also, it feels very unfinished as a discussion point. Why can things with shells be planktonic and those without not? It feels like there should at least be some speculation here, or at least a statement to say why speculation is not possible. In general, I think the structure could be improved to allow the narrative to flow more clearly.

3.As I mentioned I am not familiar with molecular methods, but the authors state the molecular divergence estimates support the emergence of extant clades after the KPg, suggesting none survived the event, but figure 4 shows otherwise (if I am reading it correctly) with both monolamellar and spinose groups originating in the Cretaceous. Even accounting for uncertainty, the suggestion is pre-KPg based on the position of the stars.

I have made a number of other more specific comments directly on the manuscript.

Reviewer #2:

Remarks to the Author:

This is an exciting study that provides conclusive evidence that our assumptions, spanning many decades, for how the planktonic foraminifera evolved and how they should be classified have been incorrect. Using environmental DNA sequencing the authors demonstrate that benthic foraminifera with diverse taxonomic affiliations are present in surface waters far offshore in the open ocean and thus they are clearly capable of a pelagic lifestyle that includes feeding and remaining buoyant. They reasonably suggest that this is likely to be a dispersal mechanism that has previously not been appreciated especially for the high number of benthic species and their taxonomic diversity. Their barcoding results demonstrate that the modern foraminifera evolved from benthic ancestors belonging to five separate phylogenetic clades and they provide robust estimates of divergence times calculated with global clocks and local molecular clocks. These dates provide a baseline for what time interval to begin searching for the likely common benthic ancestors of the different planktonic clades. The plan of study is carefully designed to make excellent use of the TARA Ocean dataset and the latest methodologies for molecular sequencing and phylogenetic analysis. I find the data presented, interpretations of results, and conclusions to be totally convincing and I cannot find any flaws in the authors' assumptions or methodologies.

This study has very broad significance and will especially be of great interest to the biological and paleontological research communities as it demonstrates a much more complex evolutionary history of the planktonic foraminifera than previously thought and it provokes interesting questions as to how and why the different clades evolved. The discussion of divergence times and likely ancestor-descendent relationships among the benthic foraminifera in the Phylogenetic placement of *Globothalamia* MOTUs is also very useful, providing an updated resource for the foraminiferal research community.

A few minor comments:

Lines 28, 204-205: could be read to imply that all Cretaceous species became extinct at the K/Pg, but that is not the case (e.g., *Guembelitria* lineage); suggest rewording to make it clear that none of those lineages survived to modern day

Line 84: replace "However" with "Nonetheless" to avoid using twice in the same paragraph

Lines 47, 152: suggest using "environmental" in place of "climatic" since climate change has not been the only cause of past biodiversity crises among the planktonic foraminifera

Line 129-130, 135-136: Need to explain why the ability to secrete mineralized shells is a critical second stepping stone toward full transition into the plankton.

Line 306: monolamellar spelled incorrectly

Line 341: buliminids should not be capitalized since it is an informal term

Throughout text: replace "K/T" with "K/Pg" since the Tertiary is no longer formally recognized in the geological time scale.

Figure 1: It is difficult for people with color blind deficiency to differentiate the color coded taxonomic groups in this figure. I suggest adding two-letter abbreviations (e.g., Tu, Mo, Gl, NS, S, Mi, Ba) as labels in Figures 1a, 1b, and 1c; The text in 1b is difficult to read; Add explanation that dashed lines separate planktic vs. benthic groupings

Reviewer #3:

Remarks to the Author:

It is an interesting question whether extant planktonic foraminiferal lineages are survivors of the Cretaceous biotic crisis or not. Although many pelagic organisms became extinct at the end Cretaceous, pelagic life including foraminifera have rapidly recovered as shown by Lowery et al. (2018 *Nature*). To answer this question, the authors need to demonstrate (1) the polyphyletic origins of

extant planktonic foraminifera and (2) their divergences were occurred after the Cretaceous. However, I am afraid that the results of the present study are insufficient to clear .

Molecular phylogenetic approaches could reveal the early evolution such as the divergences of extant planktonic foraminiferal lineages. Indeed, the previous studies (Darling et al., 1997 Mar. Micropal., Darling et al., 1999 Paleocyanog., de Vargas et al., 1999 PNAS, Ujiie and Lipps, 2009 J. Foraminif. Res., 2009), which used the partial SSU rRNA gene sequences (~1000 bp), showed that extant planktonic foraminifera are not monophyletic. However, these studies did not clarify the phylogenetic relationship between planktonic and benthic foraminifera with robust statistical supports (bootstrap and posterior provability), because the fast and various substitution rates of planktonic foraminiferal SSU rRNA gene veils deep-branching. The later studies (Ujiie et al., 2008 Mar. Micropal. and Darling et al., 2009 PNAS) succeeded to show the polyphyletic divergence of planktonic foraminifera, which are *Gallitellia* and *Dentigloborotalia*, from the benthic foraminifera with robust phylogenetic analyses. Moreover, Ujiie et al. (2008) performed the divergence time estimation and proofed that extant *Gallitellia* diverged from the ancestral benthic foraminifera in the Miocene. Comprehensive interpretation of the estimated divergence time and the triserial planktonic foraminiferal fossil records indicated that the Cretaceous and extant species were independently diverged from benthic foraminifera. I require the authors to describe these previous achievements. Then, unsolved problems are raised.

The divergences of the major planktonic foraminifera (Spinose, Non-spinose, and Microperforates) are still unknown. The origins of these groups are probably back to the Paleogene or older, but the partial SSU rRNA gene phylogenies do not supply enough information. For example, Pawlowski et al. (2013 Mar. Micropal.) and Groussin et al. (2011 Mol. Phyl. Evol.) accumulated sequence information: full SSU rRNA gene and multigene dataset, respectively, and performed to date the divergences of foraminifera. Generally, phylogenetic analyses with deep-branches are investigated using multigene or genome based data. Based on a robust phylogeny, divergence time is estimated. Otherwise, the 95% credible intervals (CIs) become large. In the divergence time estimation, the width of the 95% CI is much important rather than the estimated date. On the contrary to the standard methods, the present study used barcoding and the partial SSU rRNA gene sequences. In fact, their phylogenies failed to obtain high statistical supports at the nodes, which placed at basal position of the major planktonic foraminiferal groups. Although they employed the alternative hypotheses test (SOWAHT), the results cannot reject either monophyletic or polyphyletic hypotheses (both were $p < 0.01$). Applying unclear topology to the divergence time estimation increased the 95% CI as shown in Supplementary Material S5. These wide intervals covered over the Cretaceous (Jurassic-Eocene), and did not answer the question: whether extant planktonic foraminiferal lineages are survivors of the Cretaceous biotic crisis or not. The data of this study are insufficient to support the conclusions.

The motivation for the present study was raised by detecting benthic Globothalamea sequences in water samples. Even far from the land, these sequences were found. This is interesting result. If depth distribution of these samples is shown together with geographic distribution, it could be helpful to revise the ecological traits of Globothalamea. Few cases have been reported that Rotaliida species have plankton-phase in their lifecycle. Unfortunately, the present study tried to use the presence of Globothalamea sequences in water for different purpose: the early evolution of extant planktonic foraminifera.

The present study interpreted that biomineralization of calcite shells is associated with the ancestral benthic foraminifera becoming plankton. This hypothesis is over-speculation. As commented in the manuscript, two distinct foraminifera (Tubothalamea and Globothalamea) form biomineral calcite shells (please see Pawlowski et al., 2013 Mar. Micropal.). These groups are polyphyletic, but it seems that planktonic foraminifera have diverged only from the Globothalamea lineage.

At last, here is some comments for the materials and methods.

-Please describe water depth for sampling. In the TARA Ocean, the data of surface water is very rich, but the deeper data are not.

-In the phylogenetic analyses, some information (e.g. the number of used base pair, trimming process) were missing.

- Have the authors conducted AU test as well? The SOWAHT result was shown in the manuscript. In the case of topology test, the AU test is also a common method.
- In the divergence time estimation, there was no explanation about the local clock concept.
- In the divergence time estimation, there was no explanation about the 95% credible interval. Concerning this matter, the detail description is required in the results.
- Constraints were hired from the fossil record of only planktonic foraminifera (except for the root). Have the authors tested to apply different constraint-sets (with or without the used constraints and benthic foraminiferal fossil record)?

We are thankful to the reviewers for the time they dedicated to evaluate our work. In the following, the comments of the referees are in bold police and our responses are indicated by these symbols >>> ... <<<. We specify the line number of the changes we have made in the version of the manuscript with track change at the end of our answers when necessary.

Reviewers' comments:

Reviewer #1 (Remarks to the Author):

Upfront I would like to highlight that I can't comment too much on the veracity of the methods, as this is not my area of expertise, I come to this story exclusively from the fossil angle and have not used any of the methods employed in this paper before.

From my perspective though, the paper is a really interesting contribution and appears to use novel methods to dig deeper into understanding the complex evolutionary history of an important fossil group, and I'm all for planktonic foraminifera evolving as indicated by this molecular data. The fact that we could explain that macroperforate species can be non-spinose or spinose because they are two distinct monophyletic clades is a super interesting result.

However, I do have a major concern about the distinction between micro- and macroperforate planktonic foraminifera throughout the MS. Much of the narrative is centred around the fact that phylogenies based on fossils consider all planktonic foraminifera to be one clade. But this is not true. The existing fossil phylogeny of Aze et al. 2011 referred to by the authors includes only the macroperforate planktonic foraminifera and by omitting the qualifier "macroperforate" when referring to these phylogenies you are consistently misrepresenting it (e.g. lines 37-38, lines 147-149, lines 157-159, lines 240 -241). It is important to make the distinction clear that Aze et al. 2011 only considered the macroperforates to be one clade and not all planktonic foraminifera.

One of the main reasons the microperforates were excluded from that phylogeny was that we considered it to be true that these taxa had arisen from multiple different benthic ancestors throughout the group's history and as such could not part of what we considered to be the monophyletic macroperforates. I am happy to concede that the macroperforate could be two monophyletic groups based on this new evidence, but even if that is true, this manuscript is currently written in a way that misrepresents the previous work based exclusively on fossils.

>>> We understand the viewpoint of the reviewer that the way we present the evolution of planktonic foraminifera as interpreted by the fossil phylogenies may appear oversimplifying. This is partly due to us omitting the qualifier "macroperforate" in parts of the text where we refer to the work of Aze et al. (2011), who only assumed monophyly of the two macroperforate clades. We have corrected the statements where relevant, for example by modifying the statement on line 39 (formerly lines 37-38), not referring to Aze et al. (2011) on line 159 (formerly lines 147-149), modifying the statement on lines 167-168 (formerly lines 157-159) to not imply monophyly and highlighted on line 278 (formerly lines 240-241) that the reference applies to the macroperforate clade. Partly, this is because of the uncertain position of the microperforate clade, which has been previously often classified with other planktonic foraminifera as belonging to the same clade, but recent work considers its origin as more unresolved. We have therefore modified the text accordingly.

I do have a few other comments:

- 1. The paper is jargon heavy. Considering Nat Comms is designed to be "wide appeal" I would think the language could do with simplifying in places. I think the MS would benefit from a better outline of what the different classes represent. I had to keep referring to the paragraph lines 49-62 to remind myself of what each name meant. A "framing" figure would be super helpful to get a better visual understanding of what the groups represent in terms of generalised ecology, phylogeny and geological history.**

>>> We thank the reviewer for this useful suggestion. We have now added a new figure (now Figure 1) to recapitulate the current understanding of Foraminifera evolution at large, focussing on taxa and events discussed in the introduction. . <<<

2.The organisation and structure are confusing in places. Lines 52 – 64 feels awkward as an end to the abstract, to the extent that I wasn't sure if the authors do think there was multiple pluses from the benthos or not? Surely the abstract should be clearer than this? Lines 123 – 136 feel out of place, it seems like discussion, but not all of the results have been presented yet. Also, it feels very unfinished as a discussion point. Why can things with shells be planktonic and those without not? It feels like there should at least be some speculation here, or at least a statement to say why speculation is not possible. In general, I think the structure could be improved to allow the narrative to flow more clearly.

>>> Lines 52-64 were the end of the Introduction where the text transitions towards the first results of the study. The abstract ended on Line 30 in the initial version of the article. Following this comment, we made changes to the text to facilitate a clearer subdivision in the manuscript for the reader. It is true that the initial section in Lines 123-136 contained discussion rather than results, but we need to present this argument to explain the importance of biomineralisation to achieve the transition into the plankton, before presenting the result of the timing of such transition. We have also modified the text to make it clear that biomineralisation appears to have been the prerequisite for planktonic life among the Globothalamea, not among the foraminifera at large. The planktonic transition among the other lineages did not occur because of the absence of an adaptation to pelagic dispersal. We have now reformulated this section of the result/discussion more clearly presented in Lines 136-146 of the revised manuscript.<<<

3.As I mentioned I am not familiar with molecular methods, but the authors state the molecular divergence estimates support the emergence of extant clades after the KPg, suggesting none survived the event, but figure 4 shows otherwise (if I am reading it correctly) with both monolamellar and spinose groups originating in the Cretaceous. Even accounting for uncertainty, the suggestion is pre-KPg based on the position of the stars.

>>>The dates in figure 5 (formerly fig. 4) show the potential timing of the separation between the present planktonic clades and their nearest present benthic foraminifera relatives (for which DNA sequences exist). Such dates are necessarily always older than when the actual plankton to benthos transition happened. Also, we stress here that the K/Pg survival hypothesis implies not that the common ancestor would have to occur before the K/Pg boundary. The hypothesis for the macroperforate clades originating from *Hedbergella* implies that the divergence would have to occur before the emergence of the *Hedbergella* lineage, i.e. before 140 Ma. This latter point is now presented more explicitly in the manuscript (also in response to the comments by Reviewer #3) on lines 206-235. <<<

I have made a number of other more specific comments directly on the manuscript.

>>> We thank the reviewer for the additional comments. We report them here and provide their answers.<<<

Line 116: Just wondering where this size class info is? There's no ref so I'm guessing its from the data? This is the first time it's mentioned, what does it mean?

>>>These data are shown in the figure 2B. We have increased the writing to make it easier to read.<<<

Lines 134-136: but why? why can things with shells be planktonic and those without not? It just feels like there should at least be some speculation here, or at least a statement to say why speculation is not possible.

>>>See our response above. We do not mean to claim that only foraminifera with biomineralised shells can be planktonic. The transition occurs in two steps: first, the ability to disperse in the plankton appears, then, only among the taxa that have it, it seems that only those that calcify, rather than agglutinate, transition into the plankton. The new explanations are shown in lines 136-146 in the revised version of the manuscript.<<<

Lines 147-151: For the macroperforate foraminifera.

>>> Yes, we have answered this point above<<<

Lines 157-161: No. Only the macroperforates are considered as such in Aze et al. 2011. The microperforates are not treated in that paper, and there are a number of extant microperforates that are evidently planktonic.

>>> Indeed, we have answered this point above<<<

Lines 185-186: I like this result, I think its really interesting and would explain our spinose and non-spinose sides of the group. Shame this part of the story didn't get more air time.

>>> We have added a sentence to highlight this result further lines 188-190.<<<

Line 200: This needs unpacking as this statement is inconsistent with Fig 4 that clearly shows the global and local divergence times for the spinose forams and monolamellar as occurring before the KPg.

>>> We have answered this point above<<<

Lines 240-241: This paper only referred to the macroperforates and not all planktonic foraminifera!

>>> Indeed, we have answered this point above<<<

Reviewer #2 (Remarks to the Author):

This is an exciting study that provides conclusive evidence that our assumptions, spanning many decades, for how the planktonic foraminifera evolved and how they should be classified have been incorrect. Using environmental DNA sequencing the authors demonstrate that benthic foraminifera with diverse taxonomic affiliations are present in surface waters far offshore in the open ocean and thus they are clearly capable of a pelagic lifestyle that includes feeding and remaining bouyant. They reasonably suggest that this is likely to be a dispersal mechanism that has previously not been appreciated especially for the high number of benthic species and their taxonomic diversity. Their barcoding results deomonstrate that the modern foraminifera evolved from benthic ancestors belonging to five separate phylogenetic clades and they provide robust estimates of divergence times calculated with global clocks and local molecular clocks. These dates provide a baseline for what time interval to begin searching for the likely common benthic ancestors of the different planktonic clades.

The plan of study is carefully designed to make excellent use of the TARA Ocean dataset and the latest methodologies for molecular sequencing and phylogenetic analysis. I find the data presented, interpretations of results, and conclusions to be totally convincing and I cannot find any flaws in the authors' assumptions or methodologies.

This study has very broad significance and will especially be of great interest to the biological and paleontological research communities as it demonstrates a much more complex evolutionary history of the planktonic foraminifera than previously thought and it provokes interesting questions as to how and why the different clades evolved. The discussion of divergence times and likely ancestor-descendent relationships among the benthic foraminifera in the Phylogenetic

placement of Globothalamea MOTUs is also very useful, providing an updated resource for the foraminiferal research community.

>>> We thank the reviewer for the extremely positive appreciation of our work.<<<

A few minor comments:

Lines 28, 204-205: could be read to imply that all Cretaceous species became extinct at the K/Pg, but that is not the case (e.g., Guembelitra lineage); suggest rewording to make it clear that none of those lineages survived to modern day.

>>> We have made the modifications. lines 29 and 239. <<<

Line 84: replace “However” with “Nonetheless” to avoid using twice in the same paragraph

>>> We have made the modifications line 97.<<<

Lines 47, 152: suggest using “environmental” in place of “climatic” since climate change has not been the only cause of past biodiversity crises among the planktonic foraminifera

>>> We have made the modifications lines 54 and 162.<<<

Line 129-130, 135-136: Need to explain why the ability to secrete mineralized shells is a critical second stepping stone toward full transition into the plankton.

>>> We have reformulated this part of the discussion after the same remark of reviewer #1. In short, the textulariids that seem to be also able to disperse in the plankton need sediment particles to build their shells, and such particles are not present in plankton, preventing them to complete a full transition to planktonic lifestyle. The reformulated section is now lines 136-146.<<<

Line 306: monolamellar spelled incorrectly

>>> We have made the modification line 343.<<<

Line 341: buliminids should not be capitalized since it is an informal term

>>> We have made the modification line 381.<<<

Throughout text: replace “K/T” with “K/Pg” since the Tertiary is no longer formally recognized in the geological time scale.

>>> We have made the modification throughout the text.<<<

Figure 1: It is difficult for people with color blind deficiency to differentiate the color coded taxonomic groups in this figure. I suggest adding two-letter abbreviations (e.g., Tu, Mo, Gl, NS, S, Mi, Ba) as labels in Figures 1a, 1b, and 1c; The text in 1b is difficult to read; Add explanation that dashed lines separate planktic vs. benthic groupings

>>> We have implemented the suggestion of the referee on what is now Figure 2 when possible.<<<

Reviewer #3 (Remarks to the Author):

It is an interesting question whether extant planktonic foraminiferal lineages are survivors of the Cretaceous biotic crisis or not. Although many pelagic organisms became extinct at the end Cretaceous, pelagic life including foraminifera have rapidly recovered as shown by Lowery et al. (2018 Nature). To answer this question, the authors need to demonstrate (1) the polyphyletic

origins of extant planktonic foraminifera and (2) their divergences were occurred after the Cretaceous. However, I am afraid that the results of the present study are insufficient to clear .

>>> We fear that the referee has not fully understood the analyses we present, because neither of their two assumptions is correct. To determine that the extant planktonic foraminifera do not originate from the Cretaceous planktonic foraminifera, we need to demonstrate that the time of divergence of the extant clades from the nearest benthic relative is incompatible with the origin of the modern taxa from the known Cretaceous lineages. What do we mean? First, the phylogeny of the Cretaceous lineages is known and if any of those gave rise to the extant lineages then the last common ancestor of the extant lineages and benthic taxa would have to be older than the date of emergence of the Cretaceous lineages. This is the reasoning we use specifically for *Hedbergella*, which has been long accepted as the ancestor of the extant macroperforate taxa. Second, the referee does not seem to realize that molecular clocks cannot provide dates on transitions from benthos to plankton. The age of the last common ancestor gives the oldest possible date for the transition, but not the date itself. This is because the actual nearest benthic relative may be extinct, and we are thus unable to detect the true (younger) divergence age or because the divergence itself did not lead to a transition to the plankton yet. Strictly speaking, none of the above requires that the extant lineages are polyphyletic. They could all have been the descendants of the same post-Cretaceous planktonic expansion. However, what we believe we ought to demonstrate, and this is the substantial discovery of our study, is a mechanism that explains how is it possible that the foraminifera could transition into the plankton repeatedly.<<<<

Molecular phylogenetic approaches could reveal the early evolution such as the divergences of extant planktonic foraminiferal lineages. Indeed, the previous studies (Darling et al., 1997 Mar. Micropal., Darling et al., 1999 Paleoceanog., de Vargas et al., 1999 PNAS, Ujiie and Lipps, 2009 J. Foraminif. Res., 2009), which used the partial SSU rRNA gene sequences (~1000 bp), showed that extant planktonic foraminifera are not monophyletic. However, these studies did not clarify the phylogenetic relationship between planktonic and benthic foraminifera with robust statistical supports (bootstrap and posterior provability), because the fast and various substitution rates of planktonic foraminiferal SSU rRNA gene veils deep-branching. The later studies (Ujiie et al., 2008 Mar. Micropal. and Darling et al., 2009 PNAS) succeeded to show the polyphyletic divergence of planktonic foraminifera, which are *Gallitellia* and *Dentigloborotalia*, from the benthic foraminifera with robust. phylogenetic analyses.

>>> Although the point made by the reviewer is correct, that previous studies could not establish the exact relationships between extant planktonic and benthic foraminifera, some of the facts mentioned by the reviewer are inaccurate. De Vargas et al. (1999) does not discuss the polyphyly of planktonic foraminifera, and *Dentigloborotalia* was first sequenced in Morard et al. (2019). The publication of Darling et al. (2009) showed that the benthic genus *Bolivina* (not *Dentigloborotalia*) has a tythropelagic lifestyle, with its life cycle being in the benthos and the plankton. The fact that these studies could do robust phylogenetic inferences is because these divergences are recent, with *Bolivina* being still considered a benthic taxon, because its life cycle is not holoplanktonic. Also, we would like to repeat the argument that we stated above. Our claim that the extant planktonic foraminifera did not descend from Cretaceous lineages does not require a robust placement of the extant lineages in the global phylogeny of the foraminifera. What we require is to show that the phylogenetic position is incompatible with what would have to be the case, had the extant lineages originated from the Cretaceous ones.<<<<

Moreover, Ujiie et al. (2008) performed the divergence time estimation and proofed that extant *Gallitellia* diverged from the ancestral benthic foraminifera in the Miocene. Comprehensive interpretation of the estimated divergence time and the triserial planktonic foraminiferal fossil records indicated that the Cretaceous and extant species were independently diverged from benthic foraminifera. I require the authors to describe these previous achievements. Then, unsolved problems are raised.

>>> We agree that this study is important considering the topic of our manuscript and now present them together with the case of *Bolivina* more explicitly than before upfront in the introduction on line 38-40. <<<

The divergences of the major planktonic foraminifera (Spinose, Non-spinose, and Microperforates) are still unknown. The origins of these groups are probably back to the Paleogene or older, but the partial SSU rRNA gene phylogenies do not supply enough information. For example, Pawlowski et al. (2013 Mar. Micropal.) and Groussin et al. (2011 Mol. Phyl. Evol.) accumulated sequence information: full SSU rRNA gene and multigene dataset, respectively, and performed to date the divergences of foraminifera. Generally, phylogenetic analyses with deep-branches are investigated using multigene or genome based data. Based on a robust phylogeny, divergence time is estimated. Otherwise, the 95% credible intervals (CIs) become large. In the divergence time estimation, the width of the 95% CI is much important rather than the estimated date. On the contrary to the standard methods, the present study used barcoding and the partial SSU rRNA gene sequences. In fact, their phylogenies failed to obtain high statistical supports at the nodes, which placed at basal position of the major planktonic foraminiferal groups.

>>> We here disagree strongly with the reviewer. The large confidence intervals are due to the high heterogeneity in rate of substitution and the resulting uncertainty will not go away if longer sequences are analysed. For example, Peijnenburg et al. (2020) present a phylogenomic reconstruction including 200 genes and ~100.000 amino acid position to evaluate the timing of diversification of Pteropods in a time frame identical to ours (Jurassic to present). Despite the much larger dataset, the dating of the nodes in their phylogeny has similar uncertainty to ours (See Fig. 2: <https://www.pnas.org/doi/epdf/10.1073/pnas.1920918117>), which indicate that it is simply the nature of the methodology, which attempts to honestly take into account the existence of substitution rate heterogeneity. Further, we are absolutely aware that our dataset cannot unambiguously resolve the entire evolutionary history of the foraminifera. But this is not what is required to support our argument. Instead, we chose an approach where we explicitly test specific hypotheses (see below) rather than drawing conclusions from the phylogeny alone. We do not need to show what exactly is the sister to each clade, we only need to show that the divergences of the modern lineages from benthic relatives must have occurred later than what would have to be necessary for them to originate from the known Cretaceous planktonic lineages.<<<

Although they employed the alternative hypotheses test (SOWAHT), the results cannot reject either monophyletic or polyphyletic hypotheses (both were $p < 0.01$).

>>> We do not understand this comment. There exists no statistical test that allows rejection or acceptance of hypotheses with absolute certainty. By stating the p-value, one provides a quantitative estimate on how likely it is that the same outcome could occur by chance. $p < 0.01$ means that the chance for observing the same pattern but drawing the wrong conclusion is less than 1 in 100. But it is not zero, not even if the p was infinitesimally small. In our case, the results for our tests were $p = 0.001$ for testing the monophyly of the three main clades and $p = 0.003$ for testing the monophyly of the macroperforate foraminifera (Spinose and Non-spinose clades). This means it is ~1000 times more likely than not that our conclusion is correct. We do not understand how the reviewer can claim this result does not allow us to reject the null hypotheses?<<<

Applying unclear topology to the divergence time estimation increased the 95% CI as shown in Supplementary Material S5.

>>> We do not understand this comment either. As explained above, even a perfectly supported topology as in Peijnenburg et al. (2020) will produce large CIs. The topology we use is not unclear since all families of benthic foraminifera are respectively monophyletic, which is consistent with the results from the literature and morphology-based classification. The topology is not globally supported, which is not new since even phylogenies with the full SSU fragment could not obtain strong branch support. This means that we work with the most likely topology, but not necessarily

with the true topology. But we never claim we work with the true phylogeny and most importantly, knowing the true phylogeny of the foraminifera from the sequence data is not necessary to provide a sufficiently strong argument against the hypotheses that extant planktonic lineages originated from Cretaceous lineages. This claim only requires some topologies to be invalidated and the age constraints on the nodes to be inconsistent. The referee notes that the confidence intervals on the ages are large. This is the result of honest estimates of uncertainty combined with large heterogeneity in substitution rates in the studied gene, and most importantly, it is caused by the high heterogeneity in substitution rate in Foraminifera. It will not go away with a different topology and importantly, despite the large uncertainty, the constraints are sufficient to support our conclusions. <<<

These wide intervals covered over the Cretaceous (Jurassic-Eocene), and did not answer the question: whether extant planktonic foraminiferal lineages are survivors of the Cretaceous biotic crisis or not. The data of this study are insufficient to support the conclusions.

>>> The referee seems to be missing the point of our argument. It is not necessary to prove that the extant planktonic lineages diverged from the nearest surviving (and sequenced) benthic relative after the Cretaceous. The Cretaceous lineages are well documented and can be traced back to the early Cretaceous (some 140 Ma ago for the previously presumed ancestor of the extant lineages, *Hedbergella*). If the extant lineages originated from this ancestor, then they must have diverged from their nearest benthic relative before 140 Ma, i.e. in the Jurassic. Despite the large uncertainty, this is what the molecular clock data do support. In terms of the large uncertainty on the molecular clock estimates, we again stress that this is the result of our strategy of providing robust estimates. We deliberately choose to give as few constraints as possible on the tree to give the maximum degree of freedom to the model to estimate the date of divergence and where calibration uncertainty exists, we used dates that would allow old ages of the divergence. We choose to consider a divergence time between the Textulariids and the Rotaliida at 250 Ma instead at 200 Ma (which for example was claimed by Groussin et al. (2011)), and we chose to retain only the most reliable divergence dates between planktonic clades. In all iterations of the molecular clock that included more constrained nodes, the divergence of the extant planktonic clades were all around or after the K/Pg boundaries and with narrower CIs. We felt that to provide a robust result, we ought to show the tree with the largest possible freedom for the molecular clock model.

However, we admit that we may have been too careful by trying to under-constrain the time-calibrated tree. Therefore, we have now redone the time calibration by adding a constraint within the benthic Rotaliida (the Calcarinoidea - Nummulitoidea split), which likely took place during the Cretaceous (See revised methodology lines 385-392). As a result, the divergence dates obtained between benthic and planktonic clades are slightly younger and the CIs are slightly reduced. We have also removed the calculation of the local clock (using only single clades of planktonic foraminifera), as there were also too few constraints and these calculations did not provide any new support, nor did they invalidate any of the results already visible in the global phylogeny. We have also extended the discussion of the time-tree calibration on lines 206-235 as it was also a request of Reviewer #1. <<<

The motivation for the present study was raised by detecting benthic Globobulimina sequences in water samples. Even far from the land, these sequences were found. This is interesting result. If depth distribution of these samples is shown together with geographic distribution, it could be helpful to revise the ecological traits of Globobulimina. Few cases have been reported that Rotaliida species have plankton-phase in their lifecycle. Unfortunately, the present study tried to use the presence of Globobulimina sequences in water for different purpose: the early evolution of extant planktonic foraminifera.

>>> Indeed, the discovery that many Globobulimina (all lineages) have the ability to disperse actively in plankton was the key prerequisite for our study. It provides the mechanism on why it is possible for the foraminifera to invade the plankton repeatedly. The fact that this trait is limited to the Globobulimina explains why there was no planktonic foraminifera prior to the emergence of Globobulimina and why all known planktonic foraminifera lineages originated from the

Globothalamea. The implication for the early evolution of planktonic foraminifera is at hand and we do not understand what is “unfortunate” about it. <<<

The present study interpreted that biomineralization of calcite shells is associated with the ancestral benthic foraminifera becoming plankton. This hypothesis is over-speculation. As commented in the manuscript, two distinct foraminifera (Tubothalamea and Globothalamea) form biomineral calcite shells (please see Pawlowski et al., 2013 Mar. Micropal.). These groups are polyphyletic, but it seems that planktonic foraminifera have diverged only from the Globothalamea lineage.

>>> It seems the reviewer has not followed our argumentation correctly. We clearly state that in our model, the transition into the plankton requires two things: pelagic dispersal ability and biomineralisation. Pelagic dispersal ability is limited to the Globothalamea. Therefore, none of the other lineages of foraminifera spawned planktonic descendants. Among the Globothalamea, all lineages in the plankton have biomineralised shells. Therefore, we speculate but we believe we do not “overspeculate”, that biomineralisation is also a pre-requisite for planktonic lifestyle. Indeed, the earliest occurrence of calcifying Rotaliida in the fossil record is almost concomitant with the appearance of planktonic taxa. The reason could be extremely simple: the non-biomineralising Rotaliida build their shells by agglutinating sediment particles and such particles are not present in plankton, preventing holoplanktonic lifestyle in the absence of active biomineralization.<<<

At last, here is some comments for the materials and methods.

-Please describe water depth for sampling. In the TARA Ocean, the data of surface water is very rich, but the deeper data are not.

>>> The TARA Ocean data we use are a publicly available dataset, with all metadata provided. We feel it is not needed to list existing data again in this contribution. For the purpose of our study, it is only relevant to note that almost all of the deep chlorophyll samples are from the depth of about 100 m and we now add this information in the method part in order to clarify the origin of the data for the reader not familiar with this dataset lines 288-289.<<<

-In the phylogenetic analyses, some information (e.g. the number of used base pair, trimming process) were missing.

>>> We have provided the sequences and all intermediary files as Supplementary Material 2. All bases were used and no trimming was done.<<<

-Have the authors conducted AU test as well? The SOWAHT result was shown in the manuscript. In the case of topology test, the AU test is also a common method.

>>> There exists a plethora of tests to compare topologies, and these tests vary in their implementation in available software, the requirements in model specification and above all, their computational burden. The AU test is indeed popular because it is widely available and quick to perform but it does not mean it is better suited for our situation. Instead, we believe that SOWHAT despite its higher computational burden, was more appropriate in our case because it implements automated parameter selection such as bootstrap sampling.

Nevertheless, we ran the AU test as well and found p-value < 0.05 for the test for a monophyletic origin of planktonic foraminifera and a p-value = 0.156 for a monophyletic origin of the macroperforate planktonic foraminifera and a p-value = 0.911 for the polyphyletic (most likely) topology.

The AU and SOWHAT are thus largely returning the same results, with the p values from AU being slightly higher. This reflects our experience, the AU test is more conservative and has higher chances than SOWHAT to produce Type II statistical errors. We thus choose to retain the SOWHAT results in the manuscript, but we also cite the AU results.

-In the divergence time estimation, there was no explanation about the local clock concept.

>>> We have now removed the calculations done with the local clock concept. As described above, these added few new constraints and the results were largely consistent with the global phylogeny. The reason we used those was an attempt to circumvent the problem of substitution rate heterogeneity, by assuming the heterogeneity would be less within local clades. It was not and the local clock results returned CI that were only slightly reduced.<<<<

-In the divergence time estimation, there was no explanation about the 95% credible interval. Concerning this matter, the detail description is required in the results.

>>> We have added an extensive discussion on the reason of the CIs and their interpretation on lines 206-235.<<<<

-Constraints were hired from the fossil record of only planktonic foraminifera (except for the root). Have the authors tested to apply different constraint-sets (with or without the used constraints and benthic foraminiferal fossil record)?

>>>This is correct and as stated above, we used this strategy to not overconstrain the tree and produce divergence time estimates that are as conservative as possible. However, as also explained above, we realize that this strategy was likely too conservative and that it did not make use of a balanced selection of constraints. Therefore we have added a constraint within the benthic clades, as suggested by the reviewer, using the relatively well-constrained split of the clades Nummulitoidea and Calcarinoidea (See lines 385-392).<<<<

Reviewers' Comments:

Reviewer #1:

Remarks to the Author:

I am happy with the changes that have been made, in particular the addition of Fig 1 and the changes to Fig 5. These are great.

One very minor semantic issue, "to modern days" used on lines 30 and 244 seems like odd wording, "to the modern day" would seem a more common way to describe "now".

Reviewer #2:

Remarks to the Author:

I continue to support publication of this paper in Nature Communications because of the significance of the results. I have inserted edits and comments in the revised text and below I provide additional comments.

- The addition of Figure 1 is helpful for those who are unfamiliar with the long-term evolutionary history of foraminifera. Note that the authors should correct the misspelling of Spirillinida
- While I agree that the authors have presented convincing evidence that the modern foraminifera evolved from benthic ancestors belonging multiple separate phylogenetic clades, I disagree with the statement on lines 251-252 that "the Cretaceous planktonic foraminifera vacated the pelagic niche after the K/Pg crisis" as evidence is not presented to support that all Cretaceous lineages terminated at the K/Pg boundary. The authors have not negated compelling stratophenetic evidence, particularly from Olsson et al. (1999) Koutsoukos et al. (2014, J. Foram. Res. 44:109-142) and Arenillas et al. (2022) that:
 - o Guembelina cretacea survived the K/Pg and was ancestral to microperforate species of Parvulorugoglobigerina, Globoconusa daubjergensis, and Woodringina.
 - o Muricohedbergella (not Hedbergella as stated on line 234) is a likely ancestor of Praemurica nikolasi and perhaps EoglobigerinaBecause the guembelitriid and hedbergellid lineages terminated in the Paleogene and are not ancestral to modern lineages, it is reasonable that there is no molecular evidence that modern species had roots in the Cretaceous. I suggest the authors back off the statement that the pelagic niche was completely vacated by the planktonic foraminifera unless they can demonstrate all Cretaceous planktonic species terminated at the boundary.

Reviewer #3:

Remarks to the Author:

Dear authors,

Thank you for your feedback. I carefully read the revised MS including supplemental materials and the response letter. It seems that the authors refused to budget on the former comments. Here, I would like to add more explanation. Hopefully, the authors will understand them.

1. Weak connection between two major results.

This study provides two independent results: one was based on the analyses of barcoding sequences (~100 bp) and another was the analyses of the partial SSU rRNA gene sequences (~1000 bp). The former dataset indicates the presences of benthic foraminiferal sequences in the eDNA obtained from the upper part of water column. This implies the potential divergence from benthic to planktic foraminifers. However, there is no data and information to reinforce such possibility. Biomineralization is not requirement for being plankton. Planktic foraminifers have biomineral shells because this trait

could be descended from ancestral benthic lineages (Globothalamea). If the authors think biomineralization is "requirement" for benthic foraminifers to be plankton, please discuss what is a functional meaning and/or any ecological advantage of biomineralization in the pelagic plankton. The latter dataset (partial SSU rRNA gene sequences) was re-sampled from the database. It seems that long-branches were excluded, but the others were not described. Is there any strategy for taxonomic sampling based on the result of the former analysis with barcoding data? When I checked the fasta file (supplementary S3), all 69 OTU are the partial SSU rRNA gene region. A huge number of sequences of Globothalamea has been deposited in the public database such as the NCBI. There are also the full-length SSU rRNA gene sequences, though there are few for planktic foraminifers. A description of taxonomic sampling is a basic item in molecular phylogenetic studies.

2. Why a robust phylogeny is required for divergence time estimation.

This study focuses on the first splits of two major planktic groups (Spinose and Non-Spinose). If their divergence times were not overlapped with the occurrence of the Cretaceous Hedbergella, the hypothesis "extant planktic foraminifers are diverged after the K/Pg boundary" is highly supported. Here, I would like to mention that this molecular-based divergence time estimation is conducted on a basis of "extant" lineages. It means that such estimation presumes the presences of "extinct" lineages among "extant" lineages. Planktic foraminifers (group) are descended from a common ancestor of sister benthic lineage via putative extinct lineages. Putative extinct lineages are unknown in the estimation, but high statistic supports at nodes in a phylogeny indicate very likely sister group relationship between two lineages: planktic and planktic, or planktic or benthic. If no support, it's unknown that a putative extinct lineage is planktic or benthic.

The phylogeny shown in this study failed to show sister group relationship between planktic and benthic clades. Therefore, it's difficult to estimate what is an ancestral lineage for planktic group. Although the revised MS newly referred Peijnenburg et al. (2020, PNAS), their phylogeny was "robust". Please check Fig. 1 of their paper.

3. 95% credible interval (CI) and multi-gene analysis.

In the divergence time estimation, 95% CI means "a lineage was diverged anytime for a period of 95% CI". The estimated age is a mean or median number of 95% CI. The estimated ages (Fig. 5 and supplementary figure S4) have extremely large 95% CIs. In particular, the estimated ages of planktic foraminiferal groups and the occurrence of the Cretaceous Hedbergella were overlapped each other. Together with my comment mentioned above, these results (weak supported phylogeny and large 95% CI) indicate both possibilities that planktic foraminiferal groups were diverged from benthic lineage or from putative planktic lineage descended from the Cretaceous planktic lineage. If the authors think 95% CI is meaningless as mentioned in the response letter, please show the reason with citations.

Most studies have amplified multi-gene set to improve accuracy in phylogenetic analysis and divergence time estimation. This strategy provides two merits: shortening 95% CIs and using multiple gene models. In the case of foraminiferal study, Groussin et al. (2011, Mol.Phyl.Evol.) and Ujiie & Ishitani (2016, PlosOne) showed good examples to shorten 95% CIs (the former merit). Although Peijnenburg et al. (2020) has still wide 95% CIs, this paper is an examples of the latter merit. Peijnenburg et al. (2020) applied multiple clock models including mixed models, which are applicable to multi-gene dataset, to evaluate the divergence times of their targetd clades. Subsequently they obtained the estimated ages, all of which were fallen in the same range (age intervals). Please see Figures S2-S4 of their paper. That's why Peijnenburg et al. (2020) concluded the diversification of pteropod along with the global carbon cycle such as K/Pg and PETM. I'm afraid that the authors midunderstood how Peijnenburg et al. (2020) came to the conclusion.

Multi-gene dataset is also useful for foraminiferal phylogeny, because many lineages show different substitution rates in the same gene regions. As the authors mentioned "heterogeneity" in foraminifers, for example Spinose group (planktic foraminifers) shows fast rate in the partial SSU rRNA gene. The analysis with multi-genes, which are useful to show phylogenetic relationship, could be regulated such bias. To date, there are no effective genome data for foraminifers. I would like to encourage that the authors will do new experiment to examine a better divergence time estimation.

4. Technical questions behind.

(1) Different algorithms between phylogenetic analysis and divergence time estimation. BEAST is based on bayesian method, on the other hand, the authors used RAxML, maximum likelihood, for the phylogenetic analysis. Concerning the presences of many putative distinct lineages in this analysis, it's better to conduct a bayesian analysis for phylogeny as well.

(2) AU test and SOWHAT

AU test enables to reject hypothesis or not. The result showed alternative hypothesis was NOT rejected. This means that planktonic foraminiferal groups are MAYBE polyphyletic, considering together with the results of other previous studies.

In the response letter, the authors said "we believe that SOWHAT despite its higher computational burden, was more appropriate in our case because it implements automated parameter selection such as bootstrap sampling.". It does not explain a reason. Please show the reason with citations.

(3) Bayesian method for divergence time estimation.

In the bayesian analysis, log-likelihood scores against generation time are plotted using Tracer to confirm that the analysis is saturated. There was no description about this in the MS.

(4) Local clock

In the revised MS, a local clock was excluded. Please explain a reason. In the case of planktic foraminifers, which substitution rates are variable, it's better to exam in with local clock model as well.

(5) Test other model and sets of fossil constrains.

Have the authors conducted the divergence time estimation with different models and/or fossil constrains?

We are thankful to the reviewers for the time they dedicated to evaluate our response to the first round of revision. In the following, the comments of the referees are in bold police and our responses are indicated by these symbols >>> ... <<<. We specify the line number of the changes we have made in the version of the manuscript with track change at the end of our answers when necessary.

Reviewer #1 (Remarks to the Author):

I am happy with the changes that have been made, in particular the addition of Fig 1 and the changes to Fig 5. These are great.

>>> We are thankful for the positive evaluation and that the effort we made to supply adequate graphics were good enough in the eyes of the reviewer <<<

One very minor semantic issue, "to modern days" used on lines 30 and 244 seems like odd wording, "to the modern day" would seem a more common way to describe "now".

>>> We have made the changes lines 28 and the part on line 244 has been rewritten.<<<

Reviewer #2 (Remarks to the Author):

I continue to support publication of this paper in Nature Communications because of the significance of the results. I have inserted edits and comments in the revised text and below I provide additional comments.

>>> We are thankful for the positive evaluation as well as the additional comments that allowed us to be as accurate as possible and respect the state of the paleontological literature.<<<

• The addition of Figure 1 is helpful for those who are unfamiliar with the long-term evolutionary history of foraminifera. Note that the authors should correct the misspelling of Spirillinida

>>>The correction of Fig. 1 has been made.<<<

• While I agree that the authors have presented convincing evidence that the modern foraminifera evolved from benthic ancestors belonging multiple separate phylogenetic clades, I disagree with the statement on lines 251-252 that “the Cretaceous planktonic foraminifera vacated the pelagic niche after the K/Pg crisis” as evidence is not presented to support that all Cretaceous lineages terminated at the K/Pg boundary. The authors have not negated compelling stratophenetic evidence, particularly from Olsson et al. (1999) Koutsoukos et al. (2014, J. Foram. Res. 44:109-142) and Arenillas et al. (2022) that:

o Guembelina cretacea survived the K/Pg and was ancestral to microperforate species of Parvulorugoglobigerina, Globoconusa daubjergensis, and Woodringina.

o Muricohedbergella (not Hedbergella as stated on line 234) is a likely ancestor of Praemurica nikolasi and perhaps Eoglobigerina
Because the guembelitriid and hedbergellid lineages terminated in the Paleogene and are not ancestral to modern lineages, it is reasonable that there is no molecular evidence that modern species had roots in the Cretaceous. I suggest the authors back off the statement that the pelagic niche was completely vacated by the planktonic foraminifera unless they can demonstrate all Cretaceous planktonic species terminated at the boundary.

>>> This is a valid point and we are grateful to the reviewer for pointing out that this part of the argument could be misunderstood. It was not our intention to claim that there were no Cretaceous planktonic survivors of the K/Pg event. Instead, we wanted to express that none of such survivors was the ancestor of the present-day diversity of planktonic foraminifera. This is also how the reviewer suggests that the argument has to be modified. We have implemented this clarification in the discussion (See lines 240-258), which now more explicitly states that any that may have survived the crisis did not give rise to modern clades, which were most likely new colonizers from the benthos.<<<

Reviewer #3 (Remarks to the Author):

Dear authors,

Thank you for your feedback. I carefully read the revised MS including supplemental materials and the response letter. It seems that the authors refused to budget on the former comments. Here, I would like to add more explanation. Hopefully, the authors will understand them.

>>> We believe that we responded to all comments of the reviewer in our previous revision and it was certainly not our intention to elude any difficult aspects of the analyses we present as raised by the review. We note that the reviewer reiterates some of the arguments to which we already responded and we explain again below in detail why these have no bearing on the conclusions of our study.

However, in their review, the referee insists on some methodological aspects, which we knew would not change the conclusions, but for which we have not provided the results in the previous version of the paper. It is fair that the referee demands that we provide these analyses, and we have now implemented these in the revised version.<<<

1. Weak connection between two major results.

This study provides two independent results: one was based on the analyses of barcoding sequences (~100 bp) and another was the analyses of the partial SSU rRNA gene sequences (~1000 bp). The former dataset indicates the presences of benthic foraminiferal sequences in the eDNA obtained from the upper part of water column. This implies the potential divergence from benthic to planktic foraminifers. However, there is no data and information to reinforce such possibility.

>>> We are afraid the referee is not paraphrasing the results of our eDNA analyses correctly. What we observe is not only “presence of benthic...sequences...in the upper part of the water column”. That would be a trivial observation that could be easily explained by passive entrainment of sediment from the seafloor into the upper water during storms. Instead, what we observe is that some of the benthic taxa whose sequences are found in the upper water column show a pattern of distribution as a function of distance from the coast that is not compatible with passive entrainment and requires the capacity to remain alive and afloat for some time. This capacity is only present among the Globothalamea and because it is present in all lineages of that clade, therefore it very likely is a plesiomorphic trait that evolved early in the diversification of this group. Such capacity to stay alive in the plankton far away from the coast is not observed for the Monothalamea and Tubothalamea clades. Since all modern planktonic foraminifera lineages derived from the Globothalamea, which we can show by the analyses shown in Fig 3, and there were no transitions from the benthos into the plankton before the emergence of the Globothalamea, the only plausible conclusion is that the Globothalamea are the only clade of foraminifera which can transition into the plankton. This argument is presented

exactly in this manner on Lines 101 to 130 in the manuscript. The referee seems not to agree that such data support “the potential divergence from benthic to planktic foraminifers”. The fossil record clearly shows that benthic foraminifera existed before planktonic foraminifera, so the transfer must have occurred from the benthos into the plankton. Because all extant planktonic lineages have their nearest ancestors among the Globothalamea, they must have emerged from among benthic lineages of Globothalamea. This means there is no evidence for a divergence of benthic foraminifera from the plankton and there is no evidence for the divergence of the planktonic foraminifera from any other clade.<<<

Biom mineralization is not requirement for being plankton. Planktic foraminifers have biomineral shells because this trait could be descended from ancestral benthic lineages (Globothalamea). If the authors think biomineralization is “requirement” for benthic foraminifers to be plankton, please discuss what is a functional meaning and/or any ecological advantage of biomineralization in the pelagic plankton.

>>>Again, the referee is not paraphrasing our argument correctly. What we write on lines 134-141 of the manuscript is this:

“Since among the Globothalamea, only lineages with biomineralized shells have completed the full transition to the planktonic lifestyle, the second stepping stone on the transition into the plankton may have been the ability to secrete mineralized shells. The reason why biomineralisation would be the key to holoplanktonic lifestyle among the Globothalamea may be simple: we note that the non-biomineralising Globothalamea (textulariids) build their shells by agglutinating sediment particles and such particles are not present in the plankton, possibly preventing holoplanktonic lifestyle in the absence of complete biomineralization.”

This clearly show that we never stated that “biomineralisation is a requirement for being plankton”. Instead, we observe that among the Globothalamea foraminifera, only those who biomineralise their shells have made it into the plankton and speculate on why the ability to biomineralise would appear to be a prerequisite for the planktonic transition among those organisms. We also provide a potential explanation in the next sentence. Nowhere in the manuscript have we made a claim that biomineralisation should be a requirement to become a plankton for any other group of organisms. We believe that we must provide some explanation for the fact that all Globothalamea are found occasionally in the plankton but the non-biomineralising Textulariids have not produced any holoplanktonic clades. The Textulariids build their shells by cementing particles, which are present in the sediment of their habitat. This means they are likely unable to grow their shells in the plankton, where such material is not available.<<<

The latter dataset (partial SSU rRNA gene sequences) was re-sampled from the database. It seems that long-branches were excluded, but the others were not described. Is there any strategy for taxonomic sampling based on the result of the former analysis with barcoding data?

>>>This is a fair comment – we agree that the taxon sampling strategy for Figure 4 could have been explained more explicitly in the methods section, which we have now done. Basically, our strategy was to make sure that we represent by more than one taxon all benthic superfamilies of Rotaliida that were recognised and sampled in the work of Holzmann and Pawlowski (2017) and avoided taxa that form long branches (Lines 354-367). A complete coverage is necessary in order to attempt to place the origin of the planktonic clades in the phylogeny.<<<

When I checked the fasta file (supplementary S3), all 69 OTU are the partial SSU rRNA gene region. A huge number of sequences of Globobulimina has been deposited in the public database such as the NCBI. There are also the full-length SSU rRNA gene sequences, though there are few for planktic foraminifers. A description of taxonomic sampling is a basic item in molecular phylogenetic studies.

>>>As stated above, we have now explained the taxon selection in the methods section on lines 354-367. We now also explain the decision to analyse only the 5' fragment of the SSU RNA gene, rather than the full gene: this is because only a handful of the planktonic taxa are represented by full gene sequences and there does not exist any published sequence of a taxon from among the spinose clade.<<<

2. Why a robust phylogeny is required for divergence time estimation.

This study focuses on the first splits of two major planktic groups (Spinose and Non-Spinose). If their divergence times were not overlapped with the occurrence of the Cretaceous Hedbergella, the hypothesis “extant planktic foraminifers are diverged after the K/Pg boundary” is highly supported. Here, I would like to mention that this molecular-based divergence time estimation is conducted on a basis of “extant” lineages. It means that such estimation presumes the presences of “extinct” lineages among “extant” lineages. Planktic foraminifers (group) are descended from a common ancestor of sister benthic lineage via putative extinct lineages. Putative extinct lineages are unknown in the estimation, but high statistic supports at nodes in a phylogeny indicate very likely sister group relationship between two lineages: planktic and planktic, or planktic or benthic. If no support, it's unknown that a putative extinct lineage is planktic or benthic.

>>> This is in principle a fair point, but the referee does not seem to realise that there is absolutely no evidence for planktonic foraminifera returning back to the benthos, which means the last common ancestor of an extant planktonic lineage and its nearest relative from among the extant benthic foraminifera must have been benthic. This is because that ancestor was also an ancestor of an extant benthic clade, and if the ancestor was planktonic, there would have to be a transition from the plankton into the benthos, for which there is no evidence. Therefore, by considering the nearest benthic clade, we are actually choosing the deepest, thus oldest, possible split for the transition in the plankton. This is certainly an interesting argument which we only included implicitly and which we could indeed include more explicitly to make the reasoning stronger. This is now implemented on Line 198-209.<<<

The phylogeny shown in this study failed to show sister group relationship between planktic and benthic clades. Therefore, it's difficult to estimate what is an ancestral lineage for planktic group. Although the revised MS newly referred Peijnenburg et al. (2020, PNAS), their phylogeny was “robust”. Please check Fig. 1 of their paper.

>>>It is entirely correct that we do not resolve which of the extant benthic families is the actual ancestor of the different planktonic clades, but this has no bearing on the conclusion that the ancestor of the extant planktonic lineages could not have been the Cretaceous planktonic foraminifera. Because the phylogeny is not resolved, we test specifically for the likelihood that the planktonic lineages are monophyletic. If the phylogeny was resolved, we would not have to do that. Also, we would like to point out that we have not introduced the publication of Peijnenburg et al. (2020) to discuss the role of resolved or unresolved topology of the tree. We have referred to this work to point out that a resolved phylogeny based on multiple gene does not lead to shorter confidence intervals on the age estimates of deep splits.<<<

3. 95% credible interval (CI) and multi-gene analysis.

In the divergence time estimation, 95% CI means “a lineage was diverged anytime for a period of 95% CI”. The estimated age is a mean or median number of 95% CI. The estimated ages (Fig. 5 and supplementary figure S4) have extremely large 95% CIs. In particular, the estimated ages of planktic foraminiferal groups and the occurrence of the Cretaceous *Hedbergella* were overlapped each other. Together with my comment mentioned above, these results (weak supported phylogeny and large 95% CI) indicate both possibilities that planktic foraminiferal groups were diverged from benthic lineage or from putative planktic lineage descended from the Cretaceous planktic lineage. If the authors think 95% CI is meaningless as mentioned in the response letter, please show the reason with citations.

>>> We do not understand why the referee states that the age estimates overlap with the first emergence of *Hedbergella*. They do not. We are taking the CIs into account in our reasoning and as mentioned above, we demonstrate that the modern planktonic foraminifera cannot be descending from *Hedbergella*, as it would require that the CIs on their split from the extant benthic ancestors would have to extend beyond 140 Myrs, the date of the emergence of the first species of the lineage. The oldest split age + longest CIs does not extend beyond 140 Myrs, so it is unlikely that any of the modern foraminifera are descending from *Hedbergella*. Perhaps the referee is not realising that it is not the range, but the first emergence of the lineage which is important: once the lineage occurs in the plankton, it must have already diverged from its benthic ancestor. Unless there have been transitions from the plankton into the benthos, this must then be the youngest possible age for the ancestor of the extant planktonic and benthic lineages.<<<

Most studies have amplified multi-gene set to improve accuracy in phylogenetic analysis and divergence time estimation. This strategy provides two merits: shortening 95% CIs and using multiple gene models. In the case of foraminiferal study, Groussin et al. (2011, Mol.Phyl.Evol.) and Ujiié & Ishitani (2016, PlosOne) showed good examples to shorten 95% CIs (the former merit).

>>> This is entirely correct and we could even point out to a third element that multi-gene copy are usually used for: recruiting more informative site from the genome that would help resolving the evolutionary history of a given group. In the case of foraminifera, such multigene gene phylogenies are currently emerging, and as far as we can say, they all point in the same direction: they confirm the topology of the existing trees and confirm major evolutionary events evidenced by SSU based phylogenies. For instance, the recently published study by Sierra et al. (2022; <https://www.sciencedirect.com/science/article/abs/pii/S1055790322001592#f0010>), based on transcriptomic dataset and 199 genes show essentially the same topology and higher level classification in foraminifera as in the “old SSU-based” paper by Pawlowski and Holzmann (2013). Similarly, the recent sequencing of the COI by Macher et al. (2021; <https://www.nature.com/articles/s41598-021-01589-5>), showed that the phylogeny of SSU and COI genes are largely congruent. It is likely that such datasets will help resolve the exact position of the planktonic lineages at some point in the future. This will certainly provide new interesting insights, but the main conclusion of our study does not require the knowledge of the exact sister group for the planktonic clades.

Regarding shortening the CIs, the study by Ujiié and Ishitani (2016) indeed showed that using longer sequences or several genes does reduce CIs, as one would expect. However, they showed that the reduction is modest in comparison to the effect of the choice of the calibration dates

(See Figure 5: <https://journals.plos.org/plosone/article?id=10.1371/journal.pone.0148847>). Indeed, on their phylogeny that covers the evolutionary history of a single taxa (*Pulleniatina obliquiloculata*), the confidence interval on the age of the deepest node is reduced by 22% when multiple genes are used, but the same confidence interval is reduced by 70 % when the tree is constrained by more fossil calibration points. We entirely agree with this conclusion, which shows that the gain (shorter CIs) lies in the amount of calibration points used to date the tree rather than in using multigene datasets. The options we explored during the revisions of this paper considered either one or two constraints from the phylogeny of the benthic foraminifera only. This makes the result less dependent on the tree topology, but causes long confidence intervals. Therefore the presented solution is conservative and more robust, and because even the very long CI that we obtain do not overlap with the emergence of *Hedbergella*, we consider it safer to base the argument of the paper on the more conservative approach.<<<

Although Peijnenburg et al. (2020) has still wide 95% CIs, this paper is an examples of the latter merit. Peijnenburg et al. (2020) applied multiple clock models including mixed models, which are applicable to multi-gene dataset, to evaluate the divergence times of their targeted clades. Subsequently they obtained the estimated ages, all of which were fallen in the same range (age intervals). Please see Figures S2–S4 of their paper.

>>> We have looked into detail to the work of Peijnenburg et al. (2020) and we note that they labelled their Figure S2 as “Divergence times for major pteropod clades under different clock models were not markedly different”. Regarding the Figures S3 and S4 and similarly to the work of Ujiie and Ishitani (2016), it is the calibration dates used to constrain the time calibrated tree that causes the variability on the dating of the nodes. As they discuss it in the same supplementary material, their deepest node of interest has a range of 111.4-188.2 Ma which is similar to the range of 60-130 Ma in our tree for the divergence time of the Monolamellar foraminifera. Both CIs have a range of ~70 Ma despite Peijnenburg et al. (2020) having a multi gene dataset. <<<

That’s why Peijnenburg et al. (2020) concluded the diversification of pteropod along with the global carbon cycle such as K/Pg and PETM. I’m afraid that the authors midunderstood how Peijnenburg et al. (2020) came to the conclusion.

>>>We are afraid that the reviewer is misjudging the basis of our reasoning compared to Peijnenburg et al. (2020). We are using the time-calibrated tree to demonstrate that the modern clades cannot be descending from the *Hedberguella* lineage and thus re-interpret the fossil record evidence based on the observations made with the TARA Ocean dataset, that Globothalamea have the ability to disperse in plankton, and thus produce new holoplanktonic lineages that replaced those that disappeared at the K/Pg boundary or shortly after (See reviewer #2 comment). Peijnenburg et al. (2020), is using molecular evidences to show that the divergence time of major group of Pteropods precedes their earliest fossil record. They also discuss the resilience of the group to past climatic changes.

In short, we used time calibrated tree to show that the modern foraminifera are more recently diverged than the current interpretation of the fossil record suggest, and Peijnenburg et al. (2020) use the time calibrated tree to show that the Pteropods have a more ancient origin than suggested by previous studies. In short, we argue that Cretaceous planktonic foraminifera did not survived the K/Pg boundary while Peijnenburg et al. (2020) argue that the Pteropods survived the crisis. <<<

Multi-gene dataset is also useful for foraminiferal phylogeny, because many lineages show different substitution rates in the same gene regions. As the authors mentioned “heterogeneity” in foraminifers, for example Spinose group (planktic foraminifers) shows fast rate in the partial SSU rRNA gene. The analysis with multi-genes, which are useful to show phylogenetic relationship, could be regulated such bias. To date, there are no effective genome data for foraminifers. I would like to encourage that the authors will do new experiment to examine a better divergence time estimation.

>>> As mentioned above, the multigene dataset confirm the major evolutionary trend based on the SSU only. Adding more data does not imply that we would add more meaning, or understanding of the evolution of planktonic foraminifera. We want to stress that our manuscript is based not only on time calibrated phylogeny but also a large metabarcoding dataset assigned with a comprehensive reference dataset, and paleontological evidences.<<<

4. Technical questions behind.

(1) Different algorithms between phylogenetic analysis and divergence time estimation. BEAST is based on bayesian method, on the other hand, the authors used RAxML, maximum likelihood, for the phylogenetic analysis. Concerning the presences of many putative distinct lineages in this ansils, it’s better to conduct a baysian ansils for phylogeny as well.

>>> We initially used RAxML-ng only as in the vast majority of phylogenetic studies, it is the most likely topology that is shown and ours was compatible with earlier results from the literature. However, it is fair of the referee to require that such analyses are shown and we have now included the results for both the topology of the tree and molecular clock estimates using MrBayes (See Methods lines 371-378). The resulting topology was essentially identical with the results of RAxML: mutually monophyletic benthic and planktonic clades but without support to resolve the phylogenetic relationship between the clades and five independent origins for the planktonic foraminifera. We have added the posterior probabilities to the Figure 4, Supplementary Fig. S2, and provide the Bayesian topology as part of the supplementary Material 3. We have also performed a time calibrated tree on the Bayesian topology using the same constrain (Supplement 4). Again, we obtain qualitatively the same results that have been implemented in the text where necessary. <<<

(2) AU test and SOWHAT

AU test enables to reject hypothesis or not. The result showed alternative hypothesis was NOT rejected. This means that planktonic foraminiferal groups are MAYBE polyphyletic, considering together with the results of other previous studies.

In the response letter, the authors said “we believe that SOWHAT despite its higher computational burden, was more appropriate in our case because it implements automated parameter selection such as bootstrap sampling.” It does not explain a reason. Please show the reason with citations.

>>>As we explained in our previous reply, there is no statistical method that allows to reject completely any null hypothesis. A statistical test provides only a benchmark to measure how different is a given distribution compared to the null model. The accepted convention of $p=0.05$ means that one takes a 5% risk to wrongly reject the null hypothesis. In our case, there is only a single AU test, for the monophyly of macroperforates planktonic foraminifera, that returned a p-value of 0.15, meaning that we take a 15% risk to wrongly reject the hypothesis of the monophyly of macroperforate planktonic foraminifera. Because this test was above the threshold of $p=0.05$, we decided to test our topology (polyphyly of all clades of planktonic

foraminifera) against the null model and found $p=0.91$. This means that if we do not take a 15% risk of rejecting the hypothesis of a monophyly of macroperforate planktonic foraminifera, we would take 91% risk of rejecting their polyphyly, which is clearly a higher risk to be wrong. If we would have had a p-value of 0.5 for both test, then we would have had no argument to prefer one hypothesis against the other, but this is not the case.

Regarding the advantages of SOWHAT vs AU test, there is unfortunately no other justification that the one the authors of the SOWHAT test provided in their manuscript and that compelled them to developed this test. This is the reason we provided in our previous answer. We believe a further discussion on the merits of the two tests is not necessary in our manuscripts, because we have now, following the advice of the referee, carried out both and show the results of both.<<<

(3) Bayesian method for divergence time estimation.

In the baysian analysis, log-likelihood scores against generation time are plotted using Tracer to confirm that the analysis is saturated. There was no description about this in the MS.

>>>We do not believe such files are needed even in the supplementary material, because we state I the manuscript that the phylogeny is saturated. We attached the plot here and will let the editorial team decide if they judge that such technical file is needed at all for the manuscript.<<<

(4) Local clock

In the revised MS, a local clock was excluded. Please explain a reason. In the case of planktic foraminifers, which substitution rates are variable, it's better to examin with local clock model as well.

>>>Indeed, we initially thought that providing local clocks would help to prove our point by showing that trees with different constrains still showed the disconnection between the Cretaceous and modern planktonic foraminifera, which they did (see first submission). In the course of the first round of reviews, we realized it is a more consistent approach to provide only a global tree, where all three divergences are based on the same constraints. As we discuss below, we have explored two different sets of constraints for the global tree. Using further combinations of constraints would likely lead to further reduction of the CIs, as shown by Ujje and Ishitani (2016), but because already the conservative solution we use provides sufficiently short CI to reject the *Hedbergella* ancestry, we prefer to stick with this conservative solution for this paper. We realise that further exploration of the constraints, usage of longer gene sequences and concatenated datasets will likely allow us to pinpoint the emergence times of the clades more precisely, which will then allow us to search for their benthic ancestors in the fossil record.<<<

(5) Test other model and sets of fossil constrains.

Have the authors conducted the divergence time estimation with different models and/or fossil constrains?

>>>Yes we did. In the first version of the manuscript, we used only the divergence time of planktonic foraminifera and placed the divergence between the Textulariids and Rotaliida conservatively at 250 Ma (Instead of 200 Ma as Groussin et al. (2011). All inferences we have made provided younger divergence time with reduced CIs compared to the first version of the manuscript. We agree that the solution we have now could be further improved, but because the exact age of the divergence is not important for our argument, as long as it is younger then the emergence of *Hedbergella*, we would like to stick to the current, conservative, solution.<<<

Reviewers' Comments:

Reviewer #2:

Remarks to the Author:

Thank you for addressing the additional issues I raised and improving clarity on the issues.